# Identifying Common Hubs in Multiple Gaussian Graphical Models

**José Á. Sánchez Gómez** [1]    **Weibin Mo** [2]    **Junlong Zhao** [3]    **Yufeng Liu** [4]

## Abstract

The Gaussian graphical model (GGM) is a useful tool to represent relationships of conditional dependence among variables. In many real-world applications, datasets often contain multiple related sub-populations, whose associated GGMs may have common structure, as well as large structural differences. In such cases, it is useful to recover common hub variables, which are the highly connected variables in the GGMs of all sub-populations. In this paper, we propose the *Joint Inverse Components for Hub Detection* (JIC-HD) method to recover the common hubs across multiple GGMs without the need to estimate all subpopulation GGMs. To this end, we introduce joint minimax eigenspaces, and show that these can be leveraged for the recovery of common hubs. We establish theoretical guarantees for the recovery of common hubs. Additionally, our numerical simulation studies confirm superior performance of our JIC-HD in detecting common hubs compared to the existing methods in the literature. Our method is especially advantageous when the multiple GGMs have both common and individual hubs across sub-populations. Finally, we analyze cancer gene-expression datasets and identify biologically meaningful common hub genes across cancer subtypes.

## 1. Introduction

Research on graphical models has been prevalent in various fields including probability, machine learning and statistics the past few decades (Lauritzen, 1996; Jordan et al., 1999).

Among many approaches, the undirected Gaussian graphical model (GGM) is one of the most popular models to represent the conditional relationships among variables. For a GGM, each node in the graph represents a variable of interest, and an edge represents a conditional dependence between the corresponding variables. Thanks to its usefulness in representing variable relationships, many methods have been developed in the literature to estimate GGMs (Meinshausen & Bühlmann, 2006; Yuan & Lin, 2007; Friedman et al., 2008; Cai et al., 2011). For a recent review, see Chen (2024).

An important goal in network analysis is to identify hub nodes, which account for nodes with a substantial degree of connection compared to the rest of the nodes in the network. In a GGM, such hub nodes correspond to influential variables that have prevalent relationships with the remaining variables. Several methods have been proposed for detecting hubs in the GGM of a single population. Examples include the correlation graph screening (Hero & Rajaratnam, 2012), and the inverse covariance estimation via hub-based penalization (Tan et al., 2014; McGillivray et al., 2020). More recently, Sánchez Gómez et al. (2025) proposed a hub detection method based on spectral analysis, which leverages the low-rank structure of a GGM with hubs and avoids the direct estimation of the GGM for hub detection.

In many network applications, the population of interest may be heterogeneous, consisting of multiple sub-population graphs. For example, in cancer research on genomic data, the gene expression network can vary across sub-populations due to cancer heterogeneity (Melo et al., 2013). In particular, breast cancer can be divided into Luminal A, Luminal B, HER2-enriched and Basal-like subtypes (Yersal & Barutca, 2014). Similar subtyping has been developed for prostate cancer (Haffner et al., 2021), lung cancer (West et al., 2012), brain cancer (Verhaak et al., 2010), among others.

For data with sub-population heterogeneity, practitioners may be interested in the detection of common hubs among multiple networks, defined as those variables that are hubs across all GGMs simultaneously. For example, when analyzing gene-expression data from breast cancer patients, the expression levels of genes may be heterogeneous across different cancer subtypes. However, we expect some genes

---

[1]Department of Statistics, University of California Riverside, Riverside CA, USA [2]Department of Quantitative Methods, Daniels School of Business, Purdue University, West Lafayette IN, USA [3]Department of Statistics, Beijing Normal University, Beijing, China [4]Department of Statistics, University of Michigan, Ann Arbor MI, USA. Correspondence to: José Á. Sánchez Gómez <josesa@ucr.edu>, Weibin Mo <mo63@purdue.edu>.

*Proceedings of the 43rd International Conference on Machine Learning*, Seoul, South Korea. PMLR 306, 2026. Copyright 2026 by the author(s).

to be highly influential across all breast cancer subtypes. Recovering such common hubs can improve understanding of the gene dynamics common across all cancer subtypes, and lead to the potential development of treatments with broad effectiveness.

Despite the success of hub detection methods on a single graph, the hub discovery on multiple heterogeneous graphs remains largely unexplored. One simple strategy for common hub estimation is to estimate the set of hubs of each GGM separately regardless of their shared information, and then consider the intersection of such sets to obtain the common hubs across multiple graphs (Tan et al., 2014; McGillivray et al., 2020; Sánchez Gómez et al., 2025). However, such a strategy can suffer from the accumulation of failure rates and the inefficiency of separate modeling of multiple graphs. Alternatively, several methods have been proposed to recover common as well as individual connections across multiple GGMs (Guo et al., 2011; Mohan et al., 2014; Danaher et al., 2014; Lee & Liu, 2015; Saegusa & Shojaie, 2016; Kumar et al., 2019). More recently, transfer learning approaches have been introduced for improved graph recovery across multiple data sources (Zhao et al., 2025). These joint modeling approaches often require the individual connections not shared across the GGMs to be sparse. This assumption excludes the possibility of GGMs with non-sparse support differences, *e.g.*, if the graphical models have both common and individual hubs, where the individual hubs may be comparable, or even have a stronger signal than the common hubs.

In this paper, we propose the *joint inverse components for hub detection* (JIC-HD) method for the estimation of common hubs across multiple GGMs. Our proposed JIC-HD method is designed to identify common hubs, regardless of the presence of large support differences across the GGMs. To achieve this, we bypass the problem of support recovery that is often considered by other GGM estimation approaches. Instead we introduce a novel notion of joint minimax eigenspace across multiple GGMs, and demonstrate that these eigenspaces successfully recover the presence of common hubs across multiple graphs. As shown by our numerical studies, the proposed JIC-HD outperforms several other existing methods in terms of hub detection. We further establish the joint detection guarantee of our method under conditions of eigenspace decomposability. Additionally, we showcase the common hub-recovery performance of our proposed method on gene expression data from lung cancer patients, and successfully recover key genes associated with tumor development.

Our contributions are two-fold. First, our work is based on a novel minimax estimation approach to recover the common spectrum across multiple sub-population GGMs. We do not require the full network estimation of multiple graphs,

only focusing on the problem of common hub detection. In particular, we do not rely on the strong assumption that the multiple graphs have few differences in their network connections (Zhao et al., 2025). Second, we derive rates of convergence of joint minimax eigenspaces. Our result complements the recent studies on the minimax estimation formulation for supervised learning problems (Meinshausen & Bühlmann, 2015; Guo, 2024; Mo et al., 2024) and fair PCA (Samadi et al., 2018; Olfat & Aswani, 2019). We consider our theoretical results may be of interest to other researchers focused on the use of minimax optimization schemes to derive robust estimation of common structures across sub-populations or multiple data sources.

The rest of this paper is structured as follows. In Section 2, we introduce our notion of minimax joint eigenspaces, and show their potential for hub detection across multiple GGMs. In Section 3, we present our proposed common hub detection methodology. Theoretical guarantees of hub recovery for our proposed estimator are provided in Section 4. In Section 5, we use numerical studies to demonstrate the superior performance of our proposed method. Our analysis of gene expression data can be found in Section 6. Some discussions and future work can be found in Section 7. Proofs of the theoretical results and additional discussions are provided in the appendices.

## 2. Hubs in Gaussian Graphical Models

We first establish some notational conventions. For two non-negative sequences $\{a_p\}_{p=1}^{\infty}$ and $\{b_p\}_{p=1}^{\infty}$, we denote $a_p = O(b_p)$, or $b_p = \Omega(a_p)$, or $a_p \lesssim b_p$, if there exists a universal constant $0 < C < +\infty$ such that $a_p \leq C \cdot b_p$ for all sufficiently large $p$. The asymptotic equivalence $a_p \sim b_p$ is further defined by $a_p = O(b_p)$ and $b_p = O(a_p)$. We also denote $a_p = o(b_p)$, or $a_p \ll b_p$, or $b_p \gg a_p$, if $a_p/b_p \to 0$ as $p \to \infty$. For $K \in \mathbb{N}$, we denote $[K] := \{1, 2, \ldots, K\}$. For a finite set $A$, its cardinality is denoted as $|A|$. Let $\mathcal{S}_+^{p \times p}$ be the space of all $p \times p$ positive definite matrices. For $A, B \in \mathcal{S}_+^{p \times p}$, we denote $A \leq B$ if $B - A \in \mathcal{S}_+^{p \times p}$, $A_{\cdot k} = (A_{ik})_{i \in \mathcal{P}} \in \mathbb{R}^p$ as the vector of the $k$-th column in $A$, **rank**$(A)$ as the rank of $A$, and **span**$(A)$ as the linear subspace spanned by the columns of $A$. Let $\|\cdot\|_2$ denote the vector $\ell_2$-norm, $\|\|\cdot\|\|_2$ denote the matrix operator norm, and $\|\|\cdot\|\|_F$ denote the matrix Frobenius norm.

We consider the problem of learning the conditional dependence relationships among a set of $p$ variables $\mathcal{P} := \{1, 2, \ldots, p\}$ across $K$ given sub-populations. Suppose that $\Theta^{(1)}, \Theta^{(2)}, \ldots, \Theta^{(K)} \in \mathcal{S}_+^{p \times p}$ are positive definite precision matrices with rows and columns indexed in $\mathcal{P}$. For each $k \in [K]$, consider $\Sigma^{(k)} := (\Theta^{(k)})^{-1} \in \mathcal{S}_+^{p \times p}$ the covariance matrix associated with $\Theta^{(k)}$, and the $p$-dimensional Gaussian random vector $\boldsymbol{X}^{(k)} = \{X_i^{(k)}\}_{i \in \mathcal{P}} \sim N_p(\boldsymbol{0}, \Sigma^{(k)})$. The *Gaussian graphical model (GGM) for the distribu-*

tion of $\boldsymbol{X}^{(k)}$ is the network $\mathcal{G}^k := (\mathcal{P}, \mathcal{E}^k)$, such that for each pair of nodes $i, j \in \mathcal{P}$, $(i,j) \notin \mathcal{E}^k$ if and only if $X_i^{(k)} \perp\!\!\!\perp X_j^{(k)} \mid \boldsymbol{X}_{\setminus\{i,j\}}^{(k)} := \{X_l^{(k)} : l \in \mathcal{P}\setminus\{i,j\}\}$. In the GGM, the aforementioned conditional independence statement is equivalent to the $(i,j)$-entry of $\Theta^{(k)}$ satisfying $\Theta_{ij}^{(k)} = 0$. Thus, the sparsity in $\Theta^{(k)}$ characterizes the edge set of the GGM $\mathcal{G}^k$ (Wainwright, 2019).

## 2.1. Motivation

First, consider the problem of estimating hubs in a single GGM $\boldsymbol{X} \sim N_p(\boldsymbol{0}, \Theta^{-1})$ with associated precision matrix $\Theta \in \mathcal{S}_+^{p \times p}$. A hub variable is typically defined as the one with a high degree of connections in the associated conditional dependence graph $\mathcal{G}$, which only depends on the zero and non-zero entries of $\Theta$. With continuous elements in $\Theta$, it is relevant to account for not only the non-zeros, but also the magnitude of these entries. Based on this, we define a hub set $\mathcal{H} \subset \mathcal{P}$ as a subset of variables whose weighted degrees of connectivity are asymptotically large compared to the remaining variables, *i.e.* $\min_{h \in \mathcal{H}} \|\Theta_{\cdot h}\|_2^2 \gg \max_{i \notin \mathcal{H}} \|\Theta_{\cdot i}\|_2^2$. For the uniqueness of such a set, we further assume that the within-set degrees of connectivity are asymptotically comparable, *i.e.* $\max_{h \in \mathcal{H}} \|\Theta_{\cdot h}\|_2^2 \sim \min_{h \in \mathcal{H}} \|\Theta_{\cdot h}\|_2^2$. To characterize the signal strength of hubs, we further define the *separation rate between hubs and non-hubs* as $\tau_p := \min_{h \in \mathcal{H}} \|\Theta_{\cdot h}\|_2^2 / \max_{i \notin \mathcal{H}} \|\Theta_{\cdot i}\|_2^2$. Literature has shown that the presence of hubs in a single precision matrix $\Theta \in \mathbb{R}^{p \times p}$ implies that $\Theta$ can be approximated by a low-rank matrix $\tilde{\Theta}$, and $\tilde{\Theta}$ can be exploited for hub detection (Sánchez Gómez et al., 2025).

We motivate our proposed methodology with the example visualized in Figure 1. We consider two precision matrices $\Theta^{(1)}, \Theta^{(2)} \in \mathbb{R}^{40 \times 40}$ with hub sets $\mathcal{H}^1 = \{10, 20\}$ and $\mathcal{H}^2 = \{10, 30\}$ respectively. In this setting, there is a single common hub $\mathcal{J} = \{10\}$. Observe that, for $k = 1, 2$, the matrix $\Theta^{(k)}$ can be decomposed as $\Theta^{(k)} = \tilde{\Theta}_J + \tilde{\Theta}_{Ik} + \tilde{\Theta}_{Rk}$, where $\tilde{\Theta}_J \in \mathbb{R}^{p \times p}$ is a matrix with rank 1 that captures the presence of common hubs, $\tilde{\Theta}_{Ik} \in \mathbb{R}^{p \times p}$ represents the hub individual to the $k$-th population, and $\tilde{\Theta}_{Rk} \in \mathbb{R}^{40 \times 40}$ characterizes the remaining connections in the GGM.

From Figure 1, we observe that the recovery of the common hub $\mathcal{J} = \{10\}$ can be achieved by analyzing the common component $\Theta_J$. In particular, considering its spectral decomposition $\Theta_J = \lambda \boldsymbol{v} \boldsymbol{v}^\top$, where $\boldsymbol{v} \in \mathbb{R}^{40}$ is the eigenvector, we observe that the elements of $\boldsymbol{v}$ differentiate the common hub variable, as demonstrated in the bottom panel of Figure 1. Motivated by this observation, we develop our method based on the estimation of $\boldsymbol{v}$ via a novel notion of the joint minimax eigenspace. In this way, we bypass the need of estimating any of the graphs $\Theta^{(1)}$, $\Theta^{(2)}$, or even their common component $\Theta_J$.

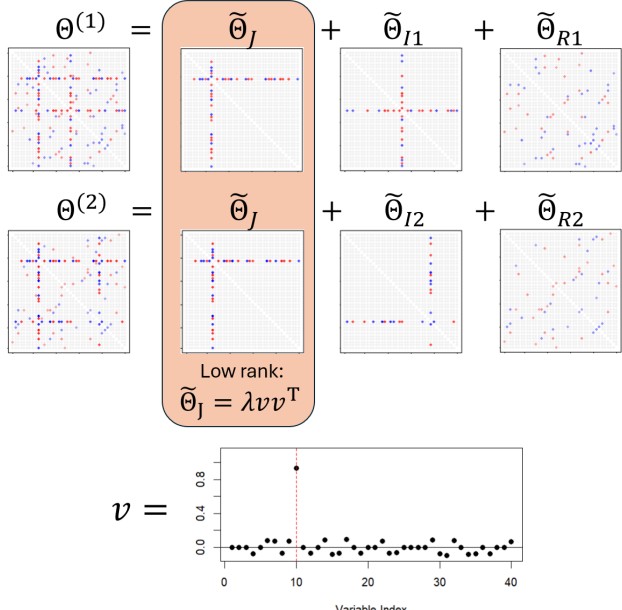

*Figure 1.* Motivating example of the presence of common low-rank structures for two precision matrices $\Theta^{(1)}, \Theta^{(2)} \in \mathbb{R}^{40 \times 40}$ with common and individual hubs. Top panels: decomposition of the precision matrices $\Theta^{(1)}$ and $\Theta^{(2)}$ into matrices containing their common hubs, individual hubs and remaining entries. The matrix containing the common hubs has rank 1, and thus may be written as $\Theta_J = \lambda \boldsymbol{v} \boldsymbol{v}^\top$. Bottom panel: Visualization of the $\boldsymbol{v} \in \mathbb{R}^{40}$. Note that $\boldsymbol{v} \in \mathbb{R}^{40}$ concentrates most of its mass on the common hub.

## 2.2. Hub Detection in Multiple GGMs

Consider a set of precision matrices $\{\Theta^{(k)}\}_{k \in [K]}$ such that, for $k \in [K]$, the matrix $\Theta^{(k)}$ contains a set of hubs $\mathcal{H}^k \subset \mathcal{P}$. The goal is to estimate the *common hub set* $\mathcal{J} := \mathcal{H}^1 \cap \mathcal{H}^2 \cap \ldots \cap \mathcal{H}^K$. We consider the following structure on the hubs of $\{\Theta^{(k)}\}_{k \in [K]}$.

**Assumption 1.** *Assume that for each $k \in [K]$, the precision matrix $\Theta^{(k)}$ contains a set of hubs $\mathcal{H}^k$, which can be represented as $\mathcal{H}^k = \mathcal{J} \cup \mathcal{I}^k$ for $\mathcal{J} \cap \mathcal{I}^k = \emptyset$, with $r := |\mathcal{J}| > 0$, $r_k := |\mathcal{I}^k| \geq 0$, and $\bigcap_{k=1}^K \mathcal{I}^k = \emptyset$. For some $\beta \in (0, 1]$ and all $k \in [K]$, assume that the hub set $\mathcal{H}^k$ of $\Theta^{(k)}$ has a hub separation rate of $\tau_p^{(k)} = \Omega(p^\beta)$.*

Assumption 1 establishes the presence of a set of common hubs $\mathcal{J} \subset \mathcal{P}$ across all populations, and allows for the presence of hub sets $\mathcal{I}^1, \mathcal{I}^2, \ldots \mathcal{I}^K$ individual to each population. Our definition of hubs in multiple graphs allows for a wide range of flexibility in the structure of the precision matrices $\{\Theta^{(k)}\}_{k \in [K]}$. By allowing for the presence of individual hubs, Assumption 1 accounts for non-sparse differences in the support of the precision matrices $\{\Theta^{(k)}\}_{k \in [K]}$. This represents an improvement from assumptions of sparse differences required by many methods of joint GGM esti-

mation (Guo et al., 2011; Mohan et al., 2014; Danaher et al., 2014; Lee & Liu, 2015; Saegusa & Shojaie, 2016; Kumar et al., 2019; Tian & Feng, 2023; Zhao et al., 2025).

The recovery of hubs depends on the hub separation rate as a signal strength. In the multiple graph setting, the hub signal strengths $\{\tau_p^{(k)}\}_{k\in[K]}$ may vary across different graphs. Assumption 1 establishes a lower bound on the minimal separation rate across all graphs, with a polynomial order $\Omega(p^\beta)$.

As discussed in Section 2.1, the presence of common hubs in $\{\Theta^{(k)}\}_{k\in[K]}$ induces a shared low-rank structure for all matrices. Motivated by this, we introduce a notion of shared eigenspaces across multiple graphs that can be leveraged for common hub detection. For a single precision matrix $\Theta$, the leading $s$ eigenvectors $V_s \in \mathbb{R}^{p\times s}$ of $\Theta$ can be identified via

$$V_s \in \underset{V\in\mathbb{R}^{p\times s};\, V^\top V=I_s}{\operatorname{argmin}} \left\{ \mathbf{tr}(V^\top \Sigma V) \right\}, \qquad (1)$$

where $\Sigma = \Theta^{-1}$, and $V_s$ minimizes the explained variance $\mathbf{tr}(V^\top \Sigma V)$. In particular, $V_s \in \mathbb{R}^{p\times s}$ spans the $s$-dimensional bottom eigenspace of $\Sigma$, which corresponds to the leading $s$ eigenspace of $\Theta$.

To define a notion of common eigenvectors across multiple graphs, a simple approach consists of calculating the pooled covariance matrix $\bar{\Sigma} = (\Sigma^{(1)} + \ldots + \Sigma^{(K)})/K \in \mathbb{R}^{p\times p}$, and minimizing (1) based on $\bar{\Sigma}$. We illustrate the failure of such additive strategy for $K = 2$ and $r = 1$ in Figure 2. As observed in Figure 2, the additive aggregation dilutes the common hub signal, leading to a high error rate. Such an instability is mainly due to the presence of individual hubs that confound the eigenspace of $\bar{\Sigma}$.

To obtain a stable eigenspace that incorporates the common hub information, we propose the max aggregation of the explained variance $\max_{k\in[K]}\{\mathbf{tr}(V^\top\Sigma^{(k)}V)\}$. As observed in Figure 2, the max aggregation can successfully capture the common hub information without being affected by the individual hubs. Motivated by such a stable aggregation, we propose the following *minimax eigenspace* for multiple GGMs.

**Definition 1** (Minimax Eigenspace). Fix $s \geq 1$. Based on the covariance matrices $\{\Sigma^{(k)}\}_{k\in[K]}$ for multiple graphs, consider the optimization problem

$$V_s^\# \in \underset{V\in\mathbb{R}^{p\times s};\, V^\top V=I_s}{\operatorname{argmin}} \left\{ \max_{k=1,\ldots,K} \mathbf{tr}(V^\top \Sigma^{(k)} V) \right\}. \qquad (2)$$

Then we define $\mathbf{span}(V_s^\#)$ as the *s-dimensional minimax joint eigenspace* for multiple GGMs. Moreover, for the $i$-th variable, we define $\omega_s^\#(i) = [V_s^\#(V_s^\#)^\top]_{ii}$ as the corresponding *minimax joint influence measure*.

Thanks to the use of a minimax optimization approach, the joint eigenspace $\mathbf{span}(V_s^\#)$ introduced in Definition 1 char-

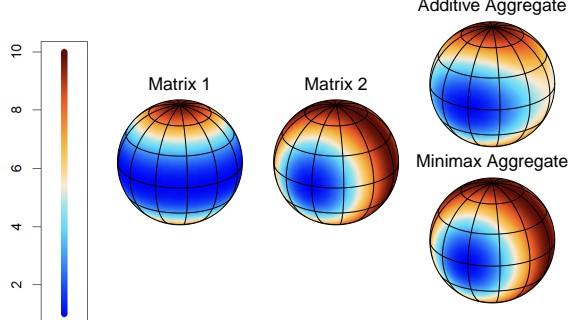

*Figure 2.* Comparison of the additive and max eigenspace problems (1). Left: Visualization of $f_1(\boldsymbol{v}) = \boldsymbol{v}^\top \Sigma^{(1)} \boldsymbol{v}$ with hub set $\mathcal{H}^1 = \{1,2\}$, $f_1$ minimizes on $\mathbf{span}(\{\boldsymbol{e}_1, \boldsymbol{e}_2\})$, $\boldsymbol{e}_1 = (1,0,0)^\top$ and $\boldsymbol{e}_2 = (0,1,0)^\top$. Center: Visualization of $f_2(\boldsymbol{v}) = \boldsymbol{v}^\top \Sigma^{(2)} \boldsymbol{v}$ with hub set $\mathcal{H}^2 = \{1\}$. $f_2$ minimizes on $\mathbf{span}(\{\boldsymbol{e}_1\})$. Top-right: Visualization of the additive aggregated $f_{\mathrm{add}} = (f_1 + f_2)/2$. Signal of common hub $\{1\}$ is diluted. Bottom-right: Visualization of the max-aggregated $f_{\max} = \max\{f_1, f_2\}$. The common hub $\{1\}$ is captured by $f_{\max}$ without being affected by individual hub $\{2\}$.

acterizes the low-rank structure that is common across the multiple precision matrices $\{\Theta^{(k)}\}_{k\in[K]}$, and thus, can be exploited for common hub detection across multiple GGMs. Furthermore, similar to the behavior of the joint eigenvector in Figure 1, if $V_s^\# = [\boldsymbol{v}_1^\#, \boldsymbol{v}_1^\#, \ldots, \boldsymbol{v}_s^\#] \in \mathbb{R}^{p\times s}$, we expect the vectors $\boldsymbol{v}_1^\#, \boldsymbol{v}_1^\#, \ldots, \boldsymbol{v}_s^\# \in \mathbb{R}^p$ to concentrate their mass on the common hub variables. To measure such concentration, we introduce the minimax joint influence measures $\omega_s^\#(i) = [V_s^\#(V_s^\#)^\top]_{ii} = \sum_{\ell=1}^s (\boldsymbol{v}_{\ell i}^\#)^2$. In the presence of common hubs, we expect $\min_{h\in\mathcal{J}} \omega_s^\#(h) \gg \max_{i\notin\mathcal{J}} \omega_s^\#(i)$. From this, the joint influence measures $\{\omega_s^\#(i)\}_{i\in\mathcal{P}}$ can be leveraged for common hub detection. We note that, due to the potential different scales of the matrices $\{\Sigma^{(k)}\}_{k\in[K]}$, one of the matrices $\Sigma^{(k_0)}$ may dominate the minimization problem (2). For applications, we suggest scaling the matrices $\{\Sigma^{(k)}\}_{k\in[K]}$ to have the same minimum eigenvalue.

The effectiveness of the minimax joint influence measures for common hub detection is illustrated in Figure 3. Here, we generate $K = 3$ precision matrices $\Theta^{(1)}, \Theta^{(2)}, \Theta^{(3)} \in \mathcal{S}_+^{100\times100}$ with the common hub set $\mathcal{J} = \{1,2,3,4,5\}$. For $k = 1,2,3$, an additional individual hub set $\mathcal{I}^k = \{20k+1, 20k+2, 20k+3\}$ is introduced. The degrees of connectivity of common hubs, individual hubs and non-hub variables to the rest of each network are approximately 20%, 40% and 5%, respectively. Due to the higher degree of connectivity of individual hubs compared to the common hubs in this example, we observe that common hubs cannot be perfectly recovered for all populations, which can result in a low rate of common hub recovery when estimating hubs for each sub-population separately. In contrast, the joint minimax influence measures exclusively concentrate

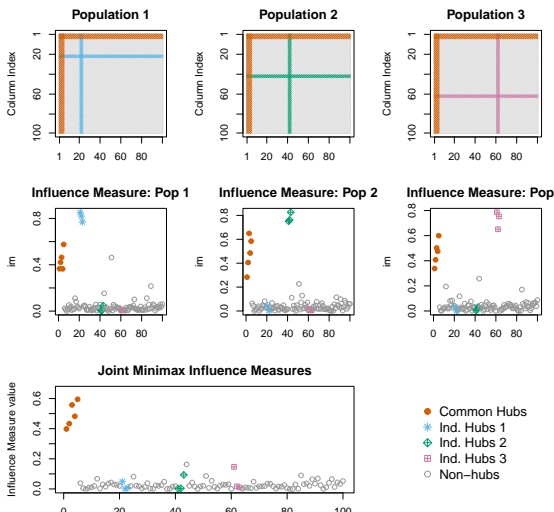

*Figure 3.* Example of the influence measures of multiple GGMs with common and individual hubs. Top panels: visualization of three precision matrices with hubs. Each matrix contains a set of common hubs, and a small set of individual hubs. Center panels: Visualization of the influence measures calculated individually. Common hubs with weaker signals may not be detected by individual hub estimation methods. Bottom panels: Plot of the joint minimax influence measures. The common hubs can be clearly detected by their joint minimax influence measures, while successfully filtering the individual hubs.

on the common hub variables, while both individual hubs and non-hub variables have low joint influence measures. This example showcases the effectiveness of the minimax joint influence measures for recovering the common hubs of multiple GGMs.

## 3. Methodology

For each $k \in [K]$, suppose that we have $n_k$ independent observations $\boldsymbol{X}_1^{(k)}, \boldsymbol{X}_2^{(k)}, \ldots, \boldsymbol{X}_{n_k}^{(k)} \sim N_p(\boldsymbol{0}, \Sigma^{(k)})$. Let $\widehat{\boldsymbol{\Sigma}}^{(k)}$ be a covariance matrix estimator associated with the data from the $k$-th sub-population. Our goal is based on the multiple-graph covariance estimators $\{\widehat{\boldsymbol{\Sigma}}^{(k)}\}_{k\in[K]}$, to estimate the common hub set $\mathcal{J}$ in Assumption 1. Further discussions on different choices of the covariance estimator and their statistical properties are provided in Section 4.

Our methodology consists of two general steps. First, we solve an empirical version of the minimax eigenspace problem (2) for $\widehat{V}_{\hat{s}}^{\#}$. Secondly, we compute the sample influence measures based on $\widehat{V}_{\hat{s}}^{\#}$. The estimated common hub set can be obtained by thesholding such sample influence measures.

We begin with the determination of the minimax eigenspace dimension $\hat{s}$ satisfying $\hat{s} \geq s$. In general, the required dimension of the minimax eigenspace $s$ is fixed and finite. Thus, a slowly growing $\hat{s}$ as $p \to \infty$ is sufficient for its

validity. In practice, we recommend the choice $\hat{s} = \sqrt{p}$, motivated from the fact that, in practice, we expect the number of hubs $r$ to satisfy $r = O(\sqrt{p})$. If the sample sizes are large enough, the estimation of $\hat{s}$ becomes feasible. We provide further details on the estimation of $s$ in the large sample size regime in Appendix C.

Upon the determination of $\hat{s}$, we solve the minimax eigenspace problem (2) with the population covariance matrices $\Sigma^{(1)}, \cdots, \Sigma^{(K)}$ replaced by their estimates $\widehat{\boldsymbol{\Sigma}}^{(1)}, \cdots, \widehat{\boldsymbol{\Sigma}}^{(K)}$. Then, with $\widehat{V}_{\hat{s}}^{\#} = [\widehat{\boldsymbol{v}}_1^{\#}, \widehat{\boldsymbol{v}}_2^{\#}, \ldots, \widehat{\boldsymbol{v}}_{\hat{s}}^{\#}] \in \mathbb{R}^{p \times \hat{s}}$, we compute the influence measures $\widehat{\omega}_{\hat{s}}(i) = [\widehat{V}_{\hat{s}}^{\#}(\widehat{V}_{\hat{s}}^{\#})^\top]_{ii} = \sum_{\ell=1}^{\hat{s}}(\widehat{\boldsymbol{v}}_{\ell i}^{\#})^2$ for all $i \in \mathcal{P}$. Based on a choice of the threshold parameter $\kappa \in (0, 1]$, we define the common hub set estimate to be $\widehat{\mathcal{J}}(\kappa) := \{h \in \mathcal{P} : \widehat{\omega}_{\hat{s}}(i) > \kappa\}$. We recommend the threshold $\hat{\kappa} = \hat{\mu}_\omega + 2\hat{\sigma}_\omega$, where $\hat{\mu}_\omega$ and $\hat{\sigma}_\omega$ are the mean and standard deviations of $\{\widehat{\omega}_{\hat{s}}(i)\}_{i\in\mathcal{P}}$, respectively. From Theorem 2, it can be shown that $\hat{\kappa}$ is of asymptotic order $O_p(\max\{p^{-\beta/4}, \sqrt{\mathcal{E}(n, p)}\})$, where $\mathcal{E}(n, p)$ measures the operator norm rate of convergence for $\{\widehat{\boldsymbol{\Sigma}}^{(k)}\}_{k=1}^K$, and that any threshold $\kappa \propto \max\{p^{-\beta/4}, \sqrt{\mathcal{E}(n, p)}\}$ is ensured to recover common hubs across multiple GGMs. This confirms the usefulness of our proposed data-driven threshold $\hat{\kappa}$.

We refer to the above method as the *Joint Inverse Components for Hub Detection (JIC-HD)*. Our proposed algorithm for common hub detection is outlined in Algorithm 1. Notice that the optimization problem (2) is constrained on a non-convex manifold. To solve (2), we apply the manifold gradient descent method (Li et al., 2021). We provide further details on the optimization algorithm in Appendix A.

---

**Algorithm 1** Joint Inverse Components for Hub Detection (JIC-HD).

---

**Inputs:** $\{\widehat{\boldsymbol{\Sigma}}^{(k)}\}_{k\in[K]}$, and $\kappa \in (0, 1]$.
Set a value of $\hat{s}$.
Solve (2) with $\Sigma^{(k)}$ replaced by $\widehat{\boldsymbol{\Sigma}}^{(k)}$ for $\widehat{V}_{\hat{s}}^{\#} \in \mathbb{R}^{p \times \hat{s}}$.
For each $i = 1, 2, \ldots, p$, compute $\widehat{\omega}_{\hat{s}}(i) = [\widehat{V}_{\hat{s}}^{\#}(\widehat{V}_{\hat{s}}^{\#})^\top]_{ii}$.
Set $\widehat{\mathcal{J}}(\kappa) = \{1 \leq i \leq p : \widehat{\omega}_{\hat{s}}(i) \geq \kappa\}$.
**Outputs:** Estimated common hub set $\widehat{\mathcal{J}}(\kappa)$.

---

## 4. Theoretical Properties

In this section, we establish theoretical guarantees of our proposed JIC-HD method. We first introduce the following assumption on the spectrum of the sub-population GGMs.

**Assumption 2** (Spike Magnitudes). *Suppose that $\{a_p\}_p$ and $\{A_p\}_p$ are some rate sequences, and for $k \in [K]$, $\lambda_1^{(k)} \geq \cdots \geq \lambda_p^{(k)}$ are the eigenvalues of the $k$-th precision matrix $\Theta^{(k)}$. Assume that for some bounded $S_k$, we have $A_p/a_p \sim p^{\beta/2}$, $\lambda_1^{(k)}, \cdots, \lambda_{S_k}^{(k)} \sim 1/a_p$, and*

$\lambda_{S_k+1}^{(k)}, \cdots, \lambda_p^{(k)} \sim 1/A_p$.

Assumption 2 serves as a characterization of the eigenvalue behavior for the precision matrices $\{\Theta^{(k)}\}_{k\in[K]}$. Specifically, it is known that a precision matrix with $r$ hubs can have $1 \le s \le r$ spiked eigenvalues (Sánchez Gómez et al., 2025). Assumption 2, ensures that both the spiked and non-spiked eigenvalues across multiple graphs share the same asymptotic orders.

Let $V_{H,k}$ denote the leading $S_k$ eigenvector matrix of $\Theta^{(k)}$. We assume that the eigenspaces $\{\mathbf{span}(V_{H,k})\}_{k\in[K]}$ are decomposable in a "common + individual" manner. For two column-orthogonal matrices $V, U \in \mathbb{R}^{p\times s}$ we denote the *chordal distance* and *chordal inner product* between their linear spans as $d_F(V, U) := \left\| VV^\top - UU^\top \right\|_F$ and $\langle V, U\rangle_F := \sqrt{\mathbf{tr}(VV^\top UU^\top)}$, respectively.

**Assumption 3** (Eigenspace Decomposability). *Consider the setting in Assumptions 1, 2, and the leading $S_k$ eigenspace* $\mathbf{span}(V_{H,k})$ *of the $k$-th precision matrix $\Theta^{(k)}$. Assume that there exists $1 \le s \le r$, such that*

$$\underbrace{\mathbf{span}(V_{H,k})}_{\dim=S_k} = \underbrace{\mathbf{span}(V_{J,k})}_{\dim=s} \oplus \underbrace{\mathbf{span}(V_{I,k})}_{\dim=S_k-s}; \quad k \in [K].$$

*Here, $V_{J,k} \in \mathbb{R}^{p\times s}, V_{I,k} \in \mathbb{R}^{p\times(S_k-s)}$ are column-orthogonal matrices such that $V_{J,k}^\top V_{I,k} = \mathbf{0}$ and $V_{H,k}V_{H,k}^\top = V_{J,k}V_{J,k}^\top + V_{I,k}V_{I,k}^\top$. Moreover, $V_{J,k}$ and $V_{I,k}$ across $k \in [K]$ relate as follows.*

*A3-1 (Common Eigenspace) For any $k,\ell \in [K]$, we have $d_F(V_{J,k}, V_{J,\ell}) \lesssim p^{-\beta/4}$;*

*A3-2 (Individual Eigenspace) Suppose that $\{\varepsilon_p\}_p$ is a rate sequence decreasing to 0. For any $k \in [K]$, there exists some $\ell \in [K]$, such that $\langle V_{I,k}, V_{I,\ell}\rangle_F \lesssim \varepsilon_p$.*

In Assumption 3, the eigenspace $\mathbf{span}(V_{H,k})$ can be decomposed into a common component $\mathbf{span}(V_{J,k})$ and an individual component $\mathbf{span}(V_{I,k})$. A3-1 ensures that the common components $\{\mathbf{span}(V_{J,k})\}_{k\in[K]}$ are close to each other in chordal distance across multiple graphs. On the other hand, A3-2 ensures that each individual component $\mathbf{span}(V_{I,k})$ is nearly orthogonal for at least another $\mathbf{span}(V_{I,\ell})$. Similar decomposability assumptions are prevalent in the context of data integration (Lock et al., 2013; Feng et al., 2018; Yi et al., 2023; Prothero et al., 2024). Based on the decomposition in Assumption 3, the minimax eigenspace $V_s^\#$ in (2) is close to all of the common eigenspaces $\{V_{J,k}\}_{k\in[K]}$, and the common hub information can be fully captured by $V_s^\#$ via the minimax influence measures.

Assumption 3 requires structural similarities across our multiple GGMs, and is satisfied in a variety of scenarios. For example, as shown in Appendix B.6, if $\{\Theta^{(k)}\}_{k\in[K]}$ satisfy $\max_{k\ne\ell} \left\| \Theta^{(k)} - \Theta^{(\ell)} \right\|_2 = O(1/A_p)$, Assumption 3 holds. While similar conditions have been previously explored (Zhao et al., 2025), requiring pairwise similarity across GGMs may be restrictive, as it discards the possibility of knowledge transfer across GGMs with large differences. Assumption 3 represents a general sufficient condition for common hub estimation, which accounts for the presence of potential large pairwise differences across the GGMs via the individual-level spectral structures $\{V_{I,k}\}_{k\in[K]}$. As we demonstrate in our theoretical results, our proposed JIC-HD can successfully recover common hubs, even in the presence of individual structures. This is further confirmed in our numerical simulations, where we show that our hub estimation techniques are effective beyond requirements of pairwise similarity.

For the JIC-HD method, the empirical minimax eigenspace problem (2) is solved with general covariance estimates $\{\widehat{\boldsymbol{\Sigma}}^{(k)}\}_{k\in[K]}$. For simplicity, we assume an equal sample size $n := n_1 = \ldots = n_K$ across multiple graphs. The following assumption controls the consistency rate of our covariance estimator.

**Assumption 4** (Covariance Estimation Guarantee). *For $k \in [K]$, consider the $k$-th covariance estimator $\widehat{\Sigma}^{(k)}$. Assume that there exists universal constants $c_1, c_2, c_3 > 0$ and some $\mathcal{E}(n,p)$, such that for any $t > 0$, we have $\left\| \widehat{\Sigma}^{(k)} - \Sigma^{(k)} \right\|_2 \le c_1 \left\| \Sigma^{(k)} \right\|_2 (\mathcal{E}(n,p) + t)$ with probability at least $1 - c_2 \exp\{-c_3 n \min\{t, t^2\}\}$.*

In the following Theorem 1, we establish a concentration inequality on the chordal distance error of the empirical minimax eigenspace problem.

**Theorem 1.** *Assume that the precision matrices $\{\Theta^{(k)}\}_{k\in[K]}$ for multiple GGMs satisfy Assumptions 1, 2, 3, and the covariance estimators $\{\widehat{\Sigma}^{(k)}\}_{k\in[K]}$ satisfy Assumption 4. Let $V_s^\#$ and $\widehat{V}_s^\#$ be the solutions to the minimax eigenspace problem (2) based on the covariance matrices $\{\Sigma^{(k)}\}_{k\in[K]}$ and $\{\widehat{\Sigma}^{(k)}\}_{k\in[K]}$, respectively. Then there exists a universal constant $c_4 > 0$, such that for any $t > 0$, with probability at least $1 - Kc_2 \exp\{-c_3 n \min\{t^2/c_1, t^4/c_1^2\}\}$, we have*

$$d_F(\widehat{V}_s^\#, V_s^\#) \le c_4 \max\left\{\varepsilon_p, p^{-\beta/4}, \sqrt{c_1\mathcal{E}(n,p)}\right\} + t.$$

Theorem 1 justifies the validity of the minimax eigenspace problem with the plug-in covariance estimates. Next, we establish the common hub detection guarantee for our JIC-HD method. For simplicity, we assume that the minimax eigenspace dimension $s$ is correctly specified.

**Theorem 2.** *Consider the setups and assumptions as in Theorem 1. Further assume that $\beta > 1/2$, and the minimax*

*eigenspace dimension $s$ is correctly specified. Then there exists a universal constant $c_5 > 0$, such that for any threshold $\kappa \geq c_5 \max\left\{\varepsilon_p, p^{-\beta/4}, \sqrt{c_1 \mathcal{E}(n,p)}\right\}$, we have*

$$\mathbb{P}\left\{\widehat{\mathcal{J}}(\kappa) \subseteq \mathcal{J}\right\} \geq 1 - K c_2 e^{-c_3 n \min\{\mathcal{E}(n,p), \mathcal{E}^2(n,p)\}}.$$

Our Theorem 2 establishes guarantees for common hub detection across multiple GGMs. In particular, Theorem 2 ensures that, for an appropriate choice of the threshold $\kappa > 0$, our estimate common hub set $\widehat{\mathcal{J}}(\kappa)$ will not have false inclusions with high probability. Additionally, as long as $\mathcal{E}(n,p) \to 0$, $n\mathcal{E}^2(n,p) \to \infty$ and $p \to \infty$, the lower requirement on the threshold $\kappa$ is vanishing, while the probability of zero false positive rate is tending to 1. In this case, any fixed threshold $\kappa$ is asymptotically valid. This provides the theoretical foundation for the proposed JIC-HD for common hub detection over multiple GGMs.

Our Theorem 2 is our most general theoretical guarantee of common hub recovery across multiple GGMs. In this general setting, we have $\widehat{\mathcal{J}}(\kappa) \subseteq \mathcal{J}$ with high probability. We explore additional sufficient conditions to guarantee that our JIC-HD achieves perfect recovery. We show that if the precision matrices $\{\Theta^{(k)}\}_{k\in[K]}$ satisfy $\max_{k \neq \ell} \left\|\left\|\Theta^{(k)} - \Theta^{(\ell)}\right\|\right\|_2 = O(1/A_p)$, $\widehat{\mathcal{J}}(\kappa) = \mathcal{J}$ with high probability, for a wide range of choices of $\kappa$. The full statement of our result, and proofs are provided in Appendix B.6.

## 5. Simulation Study

In order to establish the effectiveness of our proposed JIC-HD, we conduct several simulation studies. We perform comparative simulations for different methods found in the literature for hub detection in GGMs.

### 5.1. Data Generating Method

In order to simulate data from $K$ different Gaussian graphical models, we first generate precision matrices $\Theta^{(1)}, \Theta^{(2)}, \cdots, \Theta^{(k)} \in \mathcal{S}_+^{p\times p}$ that contain common and individual hubs. Then, for each $k \in [K]$, we generate the $k$-th sample $\boldsymbol{X}_1^{(k)}, \cdots, \boldsymbol{X}_n^{(k)} \overset{i.i.d.}{\sim} N_p(0, (\Theta^{(k)})^{-1})$ for each $k = 1, 2, \ldots, K$. To generate $\{\Theta^{(k)}\}_{k\in[K]}$, we select a dimension $p \in \{100, 200, 400\}$, a set of $r$ common hubs $\mathcal{J} \subset \mathcal{P}$, where $r \in \{5, 10, 15\}$, and $K = 3$ disjoint individual hub sets $\mathcal{I}^1, \mathcal{I}^2, \mathcal{I}^3$, each containing individual hub variables. To allow for hubs with heterogeneous degrees of connectivity, for each hub $h \in \mathcal{J}$, we select a connection probability $p_h \sim \textbf{Uniform}([p_C, p_C + 0.3])$, where $p_C \in \{0.3, 0.4, 0.5\}$. We also introduce connection probabilities for individual hubs $p_I \in \{0.3, 0.5\}$, and for non hubs $p_N = 0.05$.

For each $k = 1, 2, 3$, we generate $\Theta^{(k)} \in \mathbb{R}^{p\times p}$ as follows. First, we create an adjacency matrix $\mathrm{A}^{(k)} \in \mathbb{R}^{p\times p}$, such that: (1) the common hub nodes in $\mathcal{J}$ are connected to any other nodes with probabilities $\{p_h\}_{h\in\mathcal{J}} \overset{i.i.d.}{\sim} \textbf{Uniform}([p_C, p_C + 0.3])$; (2) the individual hub nodes in $\mathcal{I}^k$ are connected to any variables in $\mathcal{P} \setminus \mathcal{J}$ with probability $p_I$; and (3) non-hub nodes $l \in \mathcal{P} \setminus (\mathcal{J} \cup \mathcal{I}^k)$ are connected to other non-hub nodes with probability $p_N$. Then, we generate the random matrix $\tilde{\Theta}^{(k)}$ with the same sparsity pattern as $\mathrm{A}^{(k)}$, with random entries $(\tilde{\Theta}_{ij}^{(k)} | A_{ij}^{(k)} = 1) \overset{d}{=} \textbf{Uniform}([-5, -4] \cup [4, 5])$ and $\tilde{\Theta}_{ij}^{(k)} = \tilde{\Theta}_{ji}^{(k)}$. We construct the matrices ensuring that the entries of the common hubs are shared across $\{\tilde{\Theta}^{(k)}\}_{k\in[K]}$. To ensure positive definiteness, we set $\Theta^{(k)} = \tilde{\Theta}^{(k)} + (\Delta - \tilde{\lambda}_{\min})I_p$, where $\tilde{\lambda}_{\min}$ is the smallest eigenvalue of the matrices $\{\tilde{\Theta}^{(k)}\}_{k\in[K]}$, and $\Delta = 2$.

Our data generating process encompasses precision matrices $\{\Theta^{(k)}\}_{k\in[K]}$ with individual hubs, which breaks the traditional assumptions of similarity for multiple GGMs (Zhao et al., 2025). We aim to test the empirical performance of our JIC-HD method at recovering common hubs in multiple GGMs, even in settings where large structural differences are present across the multiple precision matrices.

We compare our proposed JIC-HD with the GLASSO (Friedman et al., 2008), hub weighted GLASSO (McGillivray et al., 2020) and the IPC-HD methods (Sánchez Gómez et al., 2025) on their common hub detection performance. For our proposed JIC-HD method, we select $\hat{s} = \lfloor\sqrt{p}\rfloor$, and apply our method to the sample correlation matrices. For the IPC-HD method, we set the overestimation parameter $S = \lfloor p/5 \rfloor$, and apply the method to screened correlation matrices. We apply the GLASSO and HWGLASSO on the sample correlation matrix, and select the optimal tuning parameter via the Bayesian information criteria (or BIC) (Gao et al., 2012). For all methods, we select the hubs as the set of variables 2 standard deviations above the average connectivity calculated by each method. For the GLASSO, HWGLASSO and IPC-HD methods, we select the common hub set as $\widehat{\mathcal{J}} = \bigcap_{k=1}^K \widehat{\mathcal{H}}^k$, where $\widehat{\mathcal{H}}^k$ is the estimated hub set for the $K$-th population. To evaluate the performance of the methods considered, we calculate the average F-score, precision, recall, true positve rate (TPR), false positive rate (FPR), and computational time over 100 replicates.

### 5.2. Results

In Figure 4, we find a comparison of the mean F-score obtained by each method for $r = 5$, $p_C = 0.4$ and all choices of $p$, $p_I$ and $n$. We note that the JIC-HD possesses the best performance in terms of F-score for all of the simulation scenarios considered. The advantage of our proposed method is most noticeable for $p_C = 0.4 < p_I = 0.5$, i.e., when

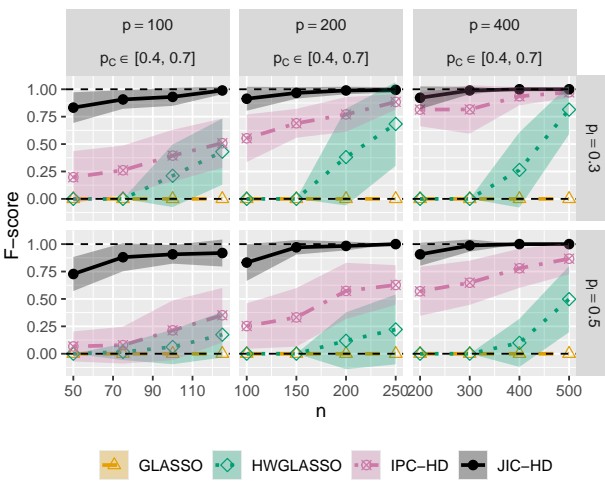

*Figure 4.* Comparison of the empirical mean F-score across methods for $p_C = 0.4$. The proposed JIC-HD provides superior performance in terms of common hub detection for all simulations provided. The improvement in common hub recovery is especially notable when individual hubs have a stronger signal than common hubs ($p_C = 0.4$, $p_I = 0.5$).

some of the common hubs may have a weaker signal than the individual hubs. In this case, other methods struggle to correctly detect the presence of the common hubs. On the other hand, our proposed method can detect the common signal, despite the presence of non-sparse differences across the graphical models.

To provide a more comprehensive understanding of the performance of our proposed JIC-HD compared to other methods in the literature, we provide further simulation results in Appendix D. First, we compare the performance in terms of common hub recovery when the number of common hubs $r$ varies to $r \in \{10, 15\}$. Additionally, we explore the sensitivity of our proposed method to the choice of $\hat{s}$, by comparing the numerical performance of our JIC-HD for $\hat{s} \in \{r, \sqrt{p}/2, \sqrt{p}, 3\sqrt{p}/2\}$. Finally, we provide additional numerical method comparisons in terms of precision, recall, true positive rate, false positive rate and computational time performance. Our additional simulation results further demonstrate the reliability and top performance of our proposed JIC-HD method for the recovery of common hubs across multiple GGMs.

## 6. Real Data Application

To exemplify the use of our JIC-HD method for common hub detection, we explore the presence of hubs in gene expression data from lung cancer patients. We sourced $n_{AD} = 540$ gene expression measurements from cancerous tissue for patients with lung adenocarcinoma (LUAD), and $n_{SC} = 511$ measurements for lung squamous cell carci-

noma patients (LUSC) from the TCGA data repository[1]. After filtering genes based on relevance to lung cancer and variance[2] and log-transforming the data (Feng et al., 2016), we analyze $p = 426$ genes across the two sub-populations. In this case, the GGM represents conditional relationships among genes. We aim to detect common hub genes, corresponding to genes with strong conditional influence for both sub-types of lung cancer. Recovering these hub genes may aid our understanding of the common biological mechanisms of both lung cancers, with potential prognostic or therapeutic benefits.

The hub detection results for our JIC-HD and the HW-GLASSO method can be found in Figure 5. Estimation results for the GLASSO and further results on HWGLASSO can be found in Appendix E. Both the JIC-HD and HW-GLASSO methods detect two clear genes as common hubs across LUAD and LUSC sub-populations: surfactant protein A 1 and 2 (SFTPA1 and SFTPA2). Both genes SFTPA1 and SFTPA2 are associated with a healthy functioning of immune response, and improved prognosis in both LUSC and LUAD (Dong et al., 2024). For example, lower SFTPA2 gene expression is found in both squamous cells and adenocarcinoma compared to baseline (Grageda et al., 2015; Cho et al., 2022; Cedzyński & Świerzko, 2024). Furthermore, deletion of the expression of both genes have shown to be associated with increased risk of lung cancer relapse, for both LUSC and LUAD (Jiang et al., 2005). We consider future studies can explore the functional association that the hub genes SFTPA1 and SFTPA2 have on the genetic pathways that induce lung cancer progression.

## 7. Discussion

In this paper, we propose JIC-HD for the detection of common hubs across multiple GGMs. Our method first solves the minimax eigenspace problem based on the covariance estimates of multiple graphs, and then performs thresholding on the corresponding influence measures. In this way, we can exploit the common structure across the GGMs while remaining robust to non-sparse support differences, such as the presence of individual hubs in the multiple networks. We establish theoretical guarantees of no false inclusions for our JIC-HD method based on the decomposability of the spiked eigenspaces with common components shared across multiple graphs. The superiority of the proposed JIC-HD is further confirmed by the simulation results. Our results suggest that JIC-HD is especially beneficial in cases where the common hubs have a weaker signal than the individual hubs present in the GGMs.

For future research, it would be of interest to explore more

---

[1]Data from: https://portal.gdc.cancer.gov/
[2]Relevant genes: https://www.malacards.org/

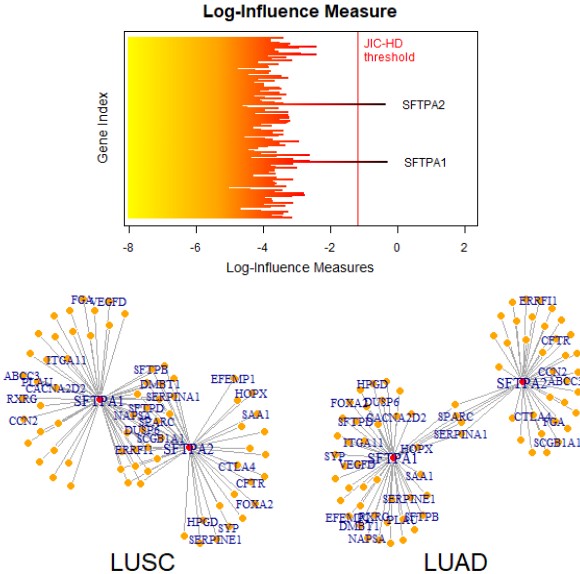

*Figure 5.* Top panel: Visualization of the JIC-HD log-influence measures for the estimation of common hub genes for LUAD and LUSC patients. SFTPA1 and SFTPA2 are detected as common hub genes. Bottom panels: Neighborhood graphs for SFTPA1 and SFTPA2 estimated via HWGLASSO for both LUAD and LUSC sub-populations.

on the minimax eigenspace problem leveraged by JIC-HD, including its statistical properties and the optimization challenge due to non-convexity.

## Impact Statement

This paper presents work whose goal is to advance the field of Machine Learning. There are many potential societal consequences of our work, none which we feel must be specifically highlighted here.

## Acknowledgements

The authors would like to thank the reviewers and area chairs, whose suggestions significantly improved the manuscript. Prof. José Á. Sánchez Gómez would like to thank the writing group of the UCR Latino and Latin American Studies Research Center, which provided a supportive environment to work on revisions to the final manuscript.

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

## A. Optimization

The most important step in the JIC-HD outlined in Algorithm 1 is the calculation of the joint eigenvectors $\widehat{V}_s^{\#}$, which requires solving (2). Notice that this optimization problem is non-convex, due to the orthonormality constraints imposed. Despite this, the feasible sets satisfy special conditions of smoothness and curvature. More specifically, the feasible sets can be seen as an embedded smooth manifolds in a Euclidean space. By exploiting the differentiable structure present in the feasible set, we can perform a modification of the gradient descent algorithm known as *manifold gradient descent* (or MGD) (Absil et al., 2009).

### A.1. Manifolds and Gradient Descent

Intuitively, an embedded smooth manifold $\mathcal{M} \subset \mathbb{R}^d$ consist of a hypersurface that may be smoothly curved, such that at each point $x$ there exists a tangent space that approximates the set locally. Although it is possible to define smooth manifolds in more general settings (Lee, 2018), we are particularly interested in manifolds that are derived as the level sets of smooth functions $H : \mathbb{R}^d \to \mathbb{R}^k$ for some $1 \le k < d$.

**Definition 2** (Embedded Manifolds). Let $H : \mathbb{R}^{d+k} \to \mathbb{R}^k$ be an infinitely differentiable function. Let $\mathcal{M} := \left\{ x \in \mathbb{R}^{d+k} : H(x) = 0 \right\}$. Assume that for all $x \in \mathcal{M}$, the differential $D_x H \in \mathbb{R}^{k \times (d+k)}$ is full rank. Then, we say the set $\mathcal{M}$ is an *embedded smooth manifold* of dimension $\mathbf{dim}(\mathcal{M}) = d$. For any $x \in \mathcal{M}$, the *tangent space to $\mathcal{M}$ at $x$* is given by $T_x \mathcal{M} = \ker(D_x H)$.

**Example 1** (Hypersurfaces). *Let $d, k > 0$, and $A \in \mathbb{R}^{(d+k) \times k}$ a matrix with rank $k$. Then, the subspace $H_\perp(A) = \left\{ x \in \mathbb{R}^{d+k} : A^\top x = 0 \right\}$ is a $d$-dimensional embedded smooth manifold, induced by the function $H_A(x) = A^\top x$. Since the derivative of this function is $D_x H_A = A^\top$, the tangent space at any point $x \in H_\perp(A)$ is the set $H_\perp(A)$ itself. This is an example in which an embedded manifold does not have any curvature.*

**Example 2** (Spheres). *Another well-known example of an embedded manifold is the $(d-1)$-dimensional sphere $S^{d-1} = \left\{ x \in \mathbb{R}^d : \|x\|_2^2 = 1 \right\}$, which is induced by the function $f(x) = \|x\|_2^2 - 1$. It is easy to show that the gradient $\nabla_x f$ is non-zero for any $x \in S^{d-1}$, which confirms that $S^{d-1}$ is an embedded smooth manifold. Since $\nabla_x f = 2x$, the tangent space at a point $x \in S^{d-1}$ is defined as $T_x S^{d-1} = \left\{ v : (2x)^\top v = 0 \right\} = \langle x \rangle^\perp$. This is a well-known example of an embedded smooth manifold with positive curvature.*

The MGD method can be applied whenever the feasible set admits the structure of an embedded differentiable manifold.

Assume we are solving the optimization problem

$$\boldsymbol{x}^* = \operatorname*{argmin}_{\boldsymbol{x} \in \mathcal{M}} F(\boldsymbol{x}),$$

where $F$ is a convex function and $\mathcal{M} \subset \mathbb{R}^d$ is an embedded smooth manifold. The core idea behind the updating step of the MGD method is visualized in Figure 6. If we are currently at the $t$-th step $\boldsymbol{x}_t \in \mathcal{M}$, we start by calculating a sub-gradient of our objective $F$ in the ambient space $\Delta_t \in \partial_{\boldsymbol{x}_t} F$. When applying the traditional gradient descent, we derive the next step by simply moving in the direction $-\varepsilon_t \Delta_t$. Due to the potential curvature present in the set $\mathcal{M}$, taking the step directly does not guarantee that such update stays in the feasible set $\mathcal{M}$. To ensure that $\boldsymbol{x}_{t+1} \in \mathcal{M}$, we first project our step $-\varepsilon_t \Delta_t$ to the tangent space of $\mathcal{M}$ at $\boldsymbol{x}_t$. This is done by applying the orthogonal projection to the tangent $T_{\boldsymbol{x}} \mathcal{M}$, which we denote by $\Pi_{\boldsymbol{x}}$. This ensures that our subgradient is restricted to directions that are approximately close to the constrained set $\mathcal{M}$. From this, we obtain the tangent direction $-\varepsilon \Delta_t^* = -\varepsilon \Pi_{\boldsymbol{x}} \Delta_t$.

Once we obtain the tangent step direction $-\varepsilon_t \Delta_t^*$, the next step in our algorithm is to project the tangent direction down to the manifold $\mathcal{M}$. This is done selecting a family of *retraction maps* $\{R_{\boldsymbol{x}}\}_{\boldsymbol{x} \in \mathcal{M}}$. For each $\boldsymbol{x}$ in the feasible set $\mathcal{M}$, the retraction $R_{\boldsymbol{x}} : T_{\boldsymbol{x}} \mathcal{M} \to \mathcal{M}$ is a function that projects any tangent direction down to the manifold $\mathcal{M}$. By applying the chosen retraction, we obtain the updated step $\boldsymbol{x}_{t+1} = R_{\boldsymbol{x}}(-\varepsilon_t \Delta_t^*)$. For more details on retractions, see Chapter 4 of Absil et al. (2009).

Such retraction maps are not unique, and several choices of feasible retraction families may be available, depending on the manifold. A common choice of a retraction is the *metric projection*, which is given by,

$$\operatorname{Proj}_{\boldsymbol{x}}(\Delta) := \operatorname*{argmin}_{\boldsymbol{y} \in \mathcal{M}} \|(\boldsymbol{x} + \Delta) - \boldsymbol{y}\|_2^2.$$

Depending on the structure of the manifold $\mathcal{M}$, there may be a closed form for this projection, which simplifies the updating steps of the MGD algorithm. A detailed description of the general MGD method can be found in Algorithm 2

---

**Algorithm 2** Manifold Gradient Descent

---

**Inputs:** Convex function $F : \mathbb{R}^d \to \mathbb{R}$, step-sizes $\{\epsilon_t\}_t$, maximum no. of iterations $T > 0$. Set $\boldsymbol{x}_0 \in \mathcal{M}$ as starting point, and $t = 0$.
**For:** $t = 1, 2, \dots, T$
$\cdots$ Find an element $\Delta_t \in \partial F(\boldsymbol{x}_t) \in \mathbb{R}^d$.
$\cdots$ Compute the tangent projection $\Delta_t^* = \Pi_{\boldsymbol{x}_t}(\Delta_t) \in T_{\boldsymbol{x}_t} \mathcal{M}$.
$\cdots$ Set the update $\boldsymbol{x}_{t+1} = R_{\boldsymbol{x}_t}[-\epsilon_t \Delta_t^*]$.
**Outputs:** $\boldsymbol{x}_T \in \mathcal{M}$.

---

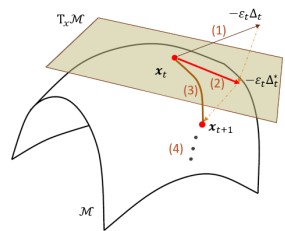

*Figure 6.* A visual representation of the MGD update step. When at a given step $\boldsymbol{x}_t$, (1) we find $\Delta_t$ a subgradient of $F$, and compute the gradient descent step for the ambient space $-\varepsilon_t \Delta_t$; (2) project the step $-\varepsilon_t \Delta_t \in \mathbb{R}^d$ to the tangent space to $\mathcal{M}$ at $\boldsymbol{x}_t$, obtaining $-\varepsilon_t \Delta_t^*$; (3) by applying a retraction $R_{\boldsymbol{x}_t}$, we derive the $(t+1)-$th step of the algorithm $\boldsymbol{x}_{t+1} = R_{\boldsymbol{x}_t}(-\varepsilon_t \Delta_t^*)$. (4) By iteratively repeating these steps, we obtain the constrained minimizer of our optimization problem.

### A.2. Gradient Descent on the Stiefel Manifold

Notice that the optimization problem (2) is constrained over the set

$$\operatorname{St}(p, s) := \left\{ V \in \mathbb{R}^{p \times s} : V^\top V = I_s \right\},$$

which is known as the $(p, s)$-Stiefel manifold. The elements $V \in \operatorname{St}(p, s)$ are known as $s$-frames.

The Stiefel manifold is an embedded manifold of dimension $d = pr - r(r+1)/2$. For any $r$-frame $X \in \operatorname{St}(p, r)$, the tangent space to $\operatorname{St}(p, r)$ at $X$ is a $d$-dimensional subspace of $\mathbb{R}^{p \times r}$ given by $T_X \operatorname{St}(p, r) := \left\{ \Delta \in \mathbb{R}^{p \times r} : X^\top \Delta + \Delta^\top X = \mathbf{0} \right\}$. Furthermore, the orthogonal projection to the tangent space $T_X \operatorname{St}(p, r)$ is given by $\Pi_X \Delta := \Delta - \frac{1}{2} X(X^\top \Delta + \Delta^\top X) \in T_X \operatorname{St}(p, r)$ for all $\Delta \in \mathbb{R}^{p \times r}$.

In order to project down the the tangent space $T_X \operatorname{St}(p, r)$ back to the manifold, we use a retraction map. We consider two retractions for the Stiefel manifold: the metric projection (Li et al., 2021) and the exponential map (Zimmermann & Huper, 2022). Given $\Delta \in T_X \operatorname{St}(p, r)$, we define the metric projection and the exponential map of $\Delta$ onto $\operatorname{St}(p, r)$ as

$$\operatorname{Proj}_X(\Delta) := \operatorname*{argmin}_{Y \in \operatorname{St}(p, r)} \|(X + \Delta) - Y\|_2^2$$
$$= (X + \Delta)(I_r + \Delta^\top \Delta)^{-1/2},$$
$$\exp_X(\Delta) := \exp_m(\Delta X^\top - X \Delta^\top) \cdot X \cdot \exp_m(-X^\top \Delta),$$

respectivly.

Given the projections and retractions defined in this section, we can apply the manifold gradient descent algorithm over the Stiefel manifold. Interested readers may explore Li et al. (2021) for further discussions.

# B. Proofs of Theoretical Results in Section 4

To show the consistency in the estimation of hubs via the joint eigenvectors $\widehat{V}_s^{\#}$, we follow a three-step proof. To begin our proof, we first provide a set of main assumptions we consider in our proof. A detailed description of our assumptions, as well as motivation for their consideration can be found in Section 4.

The first step of the proof consists of determining how the objective function $F(V) = \max_k F_k(V)$ behaves on neighborhoods of the minimizer $V_s^{\#}$. More particularly, our interest is to show that the regret $F(V) - F(V^{\#}) \geq 0$ grows quadratically with respect to the semi-distance $d_F(V, V_s^{\#})$, where, $d_F(V, U) := \left\| \| VV^T - UU^T \right\| \|_F$. The function $d_F$ is not a distance on the set of $s$-frames, since two frames have distance zero when they span the same space. In this way, the distance $d_F$ represents instead a distance between the subspaces. The tools and results for showing this step can be found in Section B.2.

The next step of our proof consists of determining the rate of convergence for the sample objective $\widehat{F}$ to the population objective function $F$ with high probability as the sample size $n, p \to \infty$. In general, we find that the uniform convergence is satisfied as long as $\mathcal{E}(n, p, t) \cdot \max_k \left\| \| \Sigma^{(k)} \right\| \|_2 \to 0$. The results concerning the probabilistic rate of convergence can be found in Section B.3

Finally, by controlling the rate of convergence of $\sup_B |\widehat{F} - F|$, and the quadratic growth of $F$ around the minimizer $V_s^{\#}$ our last step consists of deriving the rate of convergence of the sample estimator $\widehat{V}_s^{\#}$ to $V_s^{\#}$. In particular, we are interested in the rate of convergence according to the distance $d_F$. For this proof, we follow the proof structure of consistency of M-estimators found in Sen (2018). This step of the proof can be found in Section B.4.

## B.1. Equivalent Assumptions

In order to simplify our discussion, we provide the following set of equivalent assumptions, which we will be using in our proof of Theorem 1 and 2. For this, we first consider the following notation. Due to the presence of $R_k = r + r_k$ hubs in the precision matrix $\Theta^{(k)}$, we expect a number $1 \leq S_k \leq R_k$ of eigenvalues of $\Theta^{(k)}$ to separate from the rest as $p \to \infty$ (Sánchez Gómez et al., 2025, Proposition 1). To denote this, we can decompose $\Theta^{(k)}$ and $\Sigma^{(k)}$ as,

$$\Theta^{(k)} = V_{H,k} \Lambda_{H,k} V_{H,k}^\top + V_{\perp,k} \Lambda_{\perp,k} V_{\perp,k}^\top; \qquad (3)$$

$$\Sigma^{(k)} = V_{H,k} \Gamma_{H,k} V_{H,k}^\top + V_{\perp,k} \Gamma_{\perp,k} V_{\perp,k}^\top, \qquad (4)$$

where $V_{H,k} \in \mathbb{R}^{p \times S_k}$ and $V_{\perp,k} \in \mathbb{R}^{p \times (p-S_k)}$ contain the $S_k$ leading and $(p - S_k)$ bottom eigenvectors of $\Theta^{(k)}$, respectively. Furthermore, the diagonal matrices $\Lambda_{H,k} \in \mathbb{R}^{S_k \times S_k}$ and $\Lambda_{\perp,k} \in \mathbb{R}^{(p-S_k) \times (p-S_k)}$ contain the $S_k$-leading and

$(p - S_k)$-bottom eigenvalues, respectively, and $\Gamma_{H,k} = \Lambda_{H,k}^{-1}$, and $\Gamma_{\perp,k} = \Lambda_{\perp,k}^{-1}$. We introduce our Assumption 5.

**Assumption 5.** *There exists a fixed $1 < C < \infty$, and $0 < a \ll A < \infty$ such that, for each $k \in [K]$, $\mathbf{diag}(\Gamma_{\perp,k}) \subseteq [A, C \cdot A]$ and $\mathbf{diag}(\Gamma_{H,k}) \subseteq [a/C, a]$, and $A/a \sim p^{\beta/2}$.*

Assumption 5 is equivalent to our Assumption 2 from our main paper.

**Assumption 6.** *There exists a constant $1 \leq s \leq r$ such that $S_k = s + s_k$ for each $k \in [K]$. Furthermore, for each $k \in [K]$, the $S_k$-dimensional subspace $\mathbf{span}(V_{H,k}) \subset \mathbb{R}^p$ admits the orthogonal decomposition $\mathbf{span}(V_{H,k}) = \mathbf{span}(V_{J,k}) \oplus \mathbf{span}(V_{I,k})$, where,*

AS.6-1 $V_{J,k} \in \mathbb{R}^{p \times s}$ and $V_{I,k} \in \mathbb{R}^{p \times s_k}$ have orthonormal columns and $V_{J,k}^\top V_{I,k} = \mathbf{0}$;

AS.6-2 There exists a constant $M_1 > 0$ such that $\max_{k \neq l} d_F(V_{J,k}, V_{J,l}) \leq M_1 p^{-\beta/4}$;

AS.6-3 For a sequence $\{\varepsilon_{2,p}\}_p$ such that $\varepsilon_{2,p} \to 0$, we have $\max_k \min_{\ell \neq k} \left\| \| V_{I,k}^\top V_{H,\ell} \right\| \|_F \leq \varepsilon_{2,p}$.

Assumption 6 is a direct consequence of Assumption 3 from our main paper.

**Proposition 1.** *Assume the precision matrices $\{\Theta^{(k)}\}_{k \in [K]}$ satisfy Assumption 3 for a sequence $\{\varepsilon_p\}_p$. Then, there exists $C > 0$ such that Assumption 6 is satisfied, for $\varepsilon_{2,p} = C \max\{\varepsilon_p, p^{-\beta/4}\}$.*

*Proof.* If Assumption 3 holds for a sequence $\{\varepsilon_p\}_p$,

$$
\begin{aligned}
\left\| \| V_{I,k}^\top V_{H,\ell} \right\| \|_F^2 &= \left\| \| V_{I,k}^\top V_{I,\ell} \right\| \|_F^2 + \left\| \| V_{I,k}^\top V_{J,\ell} \right\| \|_F^2 \\
&= \langle V_{I,k}^\top, V_{I,\ell} \rangle_F^2 + \left\| \| V_{I,k}^\top V_{J,\ell} V_{J,\ell}^\top \right\| \|_F^2 \\
&\leq \langle V_{I,k}^\top, V_{I,\ell} \rangle_F^2 + \Big( \left\| \| V_{I,k}^\top V_{J,k} V_{J,k}^\top \right\| \|_F + \\
&\qquad \left\| \| V_{I,k}^\top (V_{J,k} V_{J,k}^\top - V_{J,\ell} V_{J,\ell}^\top) \right\| \|_F \Big)^2 \\
&\leq \langle V_{I,k}^\top, V_{I,\ell} \rangle_F^2 + d_F^2(V_{J,k}, V_{J,\ell})
\end{aligned}
$$

From this, we have that if Assumption 3 holds,

$$
\begin{aligned}
&\max_{k \in [K]} \min_{\ell \neq k} \left\| \| V_{I,k}^\top V_{H,\ell} \right\| \|_F \\
&\leq \max_{k \in [K]} \min_{\ell \neq k} \left( \langle V_{I,k}^\top, V_{I,\ell} \rangle_F^2 + d_F^2(V_{J,k}, V_{J,\ell}) \right)^{1/2} \\
&= O(\max\{\varepsilon_p, p^{-\beta/4}\}).
\end{aligned}
$$

$\qquad\qquad\qquad\qquad\qquad\qquad\qquad\qquad\qquad \square$

Moving forward, we complete the proof using Assumption 6. Then, for the conclusion of our main theorems, we apply Proposition 1 to translate our results to the notation of Assumption 3

## B.2. Quadratic Behavior of the Objective Function

### B.2.1. USEFUL TOOLS

Before we provide our results on the quadratic behavior of the objective function $F(V) = \max_k F_k(V)$, we first list some useful lemmas that will be used along our proofs.

**Proposition 2** (See Ruhe (1970)). *Let $m_1, m_2 \geq 1$, $m = \min\{m_1, m_2\}$, and $A, B \in \mathbb{R}^{m_1 \times m_2}$ be matrices with singular values $\sigma_1(A) \geq \sigma_2(A) \geq \cdots \geq \sigma_m(A) \geq 0$ and $\sigma_1(B) \geq \sigma_2(B) \geq \cdots \geq \sigma_m(B) \geq 0$, respectively. Then,*

$$\sum_{i=1}^{m} \sigma_{m-i+1}(A)\sigma_i(B) \leq |\mathbf{tr}(A^\top B)| \leq \sum_{i=1}^{m} \sigma_i(A)\sigma_i(B).$$

The usefulness of Ruhe trace inequality will be enhanced by the following proposition.

**Proposition 3.** *Let $x \in \mathbb{R}^d$, $A \in \mathbb{R}^{m_1 \times m_2}$ and $m = \min\{m_1, m_2\}$. Then, $\|x\|_1 \leq \sqrt{d}\|x\|_2$ and $\|A\|_* \leq \sqrt{m}\|A\|_F$.*

Along the following proofs, we make use of the following alternative representation of the distances between subspaces.

**Proposition 4** (Lemma 2.5 in Chen et al. (2021)). *Let $U, V \in \mathbb{R}^{p \times s}$ be matrices such that $U^\top U = V^\top V = I_s$. Furthermore, let $U_\perp, V_\perp \in \mathbb{R}^{p \times (p-s)}$ be such that $(U, U_\perp), (V, V_\perp) \in \mathcal{O}(p)$. Then,*

$$\left\|\!\left\|UU^\top - VV^\top\right\|\!\right\|_F = \sqrt{2} \cdot \left\|\!\left\|U^\top V_\perp\right\|\!\right\|_F$$
$$= \sqrt{2} \cdot \left\|\!\left\|V^\top U_\perp\right\|\!\right\|_F;$$
$$\left\|\!\left\|UU^\top - VV^\top\right\|\!\right\|_2 = \left\|\!\left\|U^\top V_\perp\right\|\!\right\|_2 = \left\|\!\left\|V^\top U_\perp\right\|\!\right\|_2.$$

From Proposition 4, we see that there is an equivalence between the proximity of two $s$-dimensional subspaces $\mathbf{span}(U)$ and $\mathbf{span}(V)$, and how close to orthogonality are $U$ and $V_\perp$. The closer the subspaces spanned by $U$ and $V$ are, the closer to orthogonality $U$ and $V_\perp$ are, so $U^\top V$ is closer to the zero matrix.

The following proposition considers the case in which two orthogonal frames $U \in \mathbb{R}^{p \times r}$ and $V \in \mathbb{R}^{p \times R}$ are almost orthogonal, *i.e.*, $\left\|\!\left\|V^\top U\right\|\!\right\|_F$ is small. In that case, we expect the frame $U$ to be closely aligned with the orthogonal complement $V_\perp$. As we show, under suitable conditions, it is possible to find an $r$-dimensional frame $U^*$ within a span contained in $\mathbf{span}(V_\perp)$ and is close to the original $U$. Intuitively, if $U$ and $V$ are close to orthogonality, we can find a frame that is simultaneously orthogonal to $V$ and close to $U$.

**Proposition 5.** *Let $U \in St(p, r)$ and $V \in St(p, R)$, where $0 < r \leq R \ll p$. Then, there exists an orthogonal $U^* \in St(p, r)$ such that $\|U - U^*\|_F \leq 2\left\|\!\left\|U^\top V\right\|\!\right\|_F$ and $\mathbf{span}(U^*) \subset \mathbf{span}(V)^\perp$.*

*Proof.* First, observe that since both $U$ and $V$ are orthogonal frames, $\left\|\!\left\|U^\top V\right\|\!\right\|_2 \leq 1$. Therefore, if $U^\top V = AQB^\top$ is the SVD decomposition of $U^\top V$, then the diagonal of $Q$ satisfies $0 \leq q_{ii} \leq 1$. Similarly, we can show that $\left\|\!\left\|U^\top VV^\top U\right\|\!\right\|_F \leq \left\|\!\left\|U^\top V\right\|\!\right\|_F$. Let $W = U - VV^\top U$. We can show that $\|W\|_2 \leq 1$. From this, given $W = XSY^\top$ the SVD decomposition of $W$ the diagonal of $S$ satisfies $0 \leq s_{ii} \leq 1$. Additionally,

$$\|U - W\|_F^2 = \left\|\!\left\|VV^\top U\right\|\!\right\|_F^2 = \mathbf{tr}(U^\top VV^\top VV^\top U)$$
$$= \mathbf{tr}(U^\top VV^\top U) = \left\|\!\left\|U^\top V\right\|\!\right\|_F^2.$$

We define $U^* = XY^\top$. By Proposition 2 it is possible to show that $U^*$ satisfies $U^* = \arg\min_{R^\top R = I_r} \|W - R\|_F^2$. Furthermore, one can show that $\mathbf{span}(U^*) = \mathbf{span}(W) \subset \mathbf{span}(V)^\perp$. Therefore, $\|U - U^*\|_F \leq \left\|\!\left\|U^\top V\right\|\!\right\|_F + \|W - U^*\|_F$.

To conclude, it suffices to show that $\|W - U^*\|_F \leq \left\|\!\left\|U^\top V\right\|\!\right\|_F$. First, observe that,

$$\|W - U^*\|_F^2 = \left\|\!\left\|X(S - I_r)Y^\top\right\|\!\right\|_F^2 = \|I_r - S\|_F^2.$$

Furthermore, $YS^2Y^\top = W^\top W = I_r - U^\top VV^\top U$. From this, it follows that $I_r - S^2 = Y^\top U^\top VV^\top UY$. Since $\mathbf{0} \leq S \leq I_r$, it follows that $|s_{ii} - 1| \leq |s_{ii}^2 - 1| \leq 1$. Therefore, $\|I_r - S\|_F^2 \leq \left\|\!\left\|I_r - S^2\right\|\!\right\|_F^2 = \left\|\!\left\|Y^\top U^\top VV^\top UY\right\|\!\right\|_F^2 = \left\|\!\left\|U^\top VV^\top U\right\|\!\right\|_F^2$. Since $\left\|\!\left\|U^\top V\right\|\!\right\|_2 \leq 1$, we conclude $\left\|\!\left\|U^\top VV^\top U\right\|\!\right\|_F^2 \leq \left\|\!\left\|U^\top V\right\|\!\right\|_F^2$. $\square$

### B.2.2. SHOWING QUADRATIC BEHAVIOR

With the propositions outlined in Section B.2.1, we are ready to show the quadratic behavior of the objective function around the minimizer $V_s^\#$. To start, we bound the value of the objective function at the minimizer.

**Lemma 1.** *Let $\{\Theta^{(k)}\}_{k \in [K]}$ be a set of precision matrices that satisfy Assumptions 1, 2 and 6. Then, there exists a constant $c_1^* > 0$ such that, for sufficiently large $p$,*

$$F(V_s^\#) = \max_{k \in [K]} F_k(V_s^\#) \leq c_1^* \cdot a.$$

*Proof.* To demonstrate the upper bound on the minimizer $F(V_s^\#) = \max_l F_l(V_s^\#) \leq M \cdot a$, we show that $F(V_{J,k}) \leq c_1^* \cdot a$ for any $k \in [K]$, where $\{V_{J,k}\}_k$ are defined as in Assumption 6. To show this, we first prove that $F_k(V_{J,k}) \leq c_1^* \cdot a$. Then, we show that for any $\ell \neq k$, $F_\ell(V_{J,k}) \leq c_1^* \cdot a$.

Let $k \in [K]$ be fixed. Since $V_{J,k}$ and $V_{H,k}$ are both orthogonal frames, it can be shown that $\left\|\!\left\|V_{H,k}^\top V_{J,k}V_{J,k}^\top V_{H,k}\right\|\!\right\|_2 \leq 1$. Furthermore, $V_{J,k}^\top V_{\perp,k} = \mathbf{0}$. By Assumption 2, we know

that $\||\Gamma_{H,k}\||_2 \leq a$. Therefore, by decomposing $\Sigma^{(k)}$ as in (3) and applying Proposition 2,

$$
\begin{aligned}
F_k(\mathrm{V}_{J,k}) &= \mathbf{tr}(\mathrm{V}_{J,k}^\top \Sigma^{(k)} \mathrm{V}_{J,k}) \\
&= \mathbf{tr}(\mathrm{V}_{H,k}^\top \mathrm{V}_{J,k} \mathrm{V}_{J,k}^\top \mathrm{V}_{H,k} \cdot \Gamma_{H,k}) \\
&\leq \sum_{i=1}^{s} \sigma_i(\mathrm{V}_{H,k}^\top \mathrm{V}_{J,k} \mathrm{V}_{J,k}^\top \mathrm{V}_{H,k}) \cdot \||\Gamma_{H,k}\||_2 \\
&\leq sa.
\end{aligned}
$$

Now, let us bound $F_\ell(\mathrm{V}_{J,k})$ for $\ell \neq k$. For this, we show that the value $F_\ell(\mathrm{V}_{J,k})$ is close to the value of $F_\ell(\mathrm{V}_{J,\ell})$. First, observe that by the decomposition (3),

$$
\begin{aligned}
F_\ell(\mathrm{V}_{J,k}) &\leq F_\ell(\mathrm{V}_{J,\ell}) + |\mathbf{tr}([\mathrm{V}_{J,k}\mathrm{V}_{J,k}^\top - \mathrm{V}_{J,\ell}\mathrm{V}_{J,\ell}^\top]\Sigma^{(\ell)})| \\
&\leq sa + |\mathbf{tr}([\mathrm{V}_{J,k}\mathrm{V}_{J,k}^\top - \mathrm{V}_{J,\ell}\mathrm{V}_{J,\ell}^\top] \cdot \mathrm{V}_{H,\ell} \cdot \Gamma_{H,\ell} \mathrm{V}_{H,\ell}^\top)| + \\
&\qquad |\mathbf{tr}(\mathrm{V}_{J,k}\mathrm{V}_{J,k}^\top \mathrm{V}_{\perp,\ell} \cdot \Gamma_{\perp,\ell} \mathrm{V}_{\perp,\ell}^\top)| \\
&\leq sa + a \cdot d_F(\mathrm{V}_{J,k}, \mathrm{V}_{J,\ell}) + CA\||\mathrm{V}_{J,k}^\top[\mathrm{V}_{\perp,\ell}, \mathrm{V}_{I,\ell}]\||_F^2 \\
&\leq sa + a \cdot d_F(\mathrm{V}_{J,k}, \mathrm{V}_{J,\ell}) + a\left[\frac{CA}{a}d_F^2(\mathrm{V}_{J,k}, \mathrm{V}_{J,\ell})\right].
\end{aligned}
$$

By Assumption 2 and 6, we have that $A/a = O(p^{\beta/2})$, $d_F(\mathrm{V}_{J,k}, \mathrm{V}_{J,\ell}) = O(p^{\beta/4})$, and therefore $\frac{A}{a}d_F^2(\mathrm{V}_{J,k}, \mathrm{V}_{J,\ell}) = O(1)$. From this, we conclude. $\qquad\square$

In Proposition 1 we provide a bound on the objective function at the minimizer $\mathrm{V}_s^\#$. The next step in our proof is to provide a quadratic lower bound on the value of the function $F(\mathrm{V})$ for $\mathrm{V} \neq \mathrm{V}_s^\#$.

**Lemma 2.** *Consider $\{\Theta^{(k)}\}_{k\in[K]}$ that satisfy Assumptions 1, 2 and 6. There exist constants $c_2^*, c_3^* > 0$ such that, for all s-frames $\mathrm{V} \in \mathrm{St}(p, s)$ such that $\max_k d_F(\mathrm{V}, \mathrm{V}_{J,k}) \geq c_2^* \varepsilon_{2,p}$. Then,*

$$
F(\mathrm{V}) \geq c_3^* \max_k \||\Sigma^{(k)}\||_2 \cdot \max_k d_F^2(\mathrm{V}, \mathrm{V}_{J,k}).
$$

*Proof.* Let $\mathrm{V} \in \mathrm{St}(p, s)$. Since $\Theta^{(k)}$ and $\Sigma^{(k)}$ satisfy Assumption 2, we know that $\Gamma_{\perp,k} \geq A \cdot \mathrm{I}_{p-S_k}$. From this, decomposing $\Sigma^{(k)}$ as in (3),

$$
\begin{aligned}
F_k(\mathrm{V}) &= \mathbf{tr}(\mathrm{V}^\top \Sigma^{(k)} \mathrm{V}) \\
&= \mathbf{tr}(\mathrm{V}^\top \mathrm{V}_{H,k} \Gamma_{H,k} \mathrm{V}_{H,k}^\top \mathrm{V}) + \mathbf{tr}(\mathrm{V}^\top \mathrm{V}_{\perp,k} \Gamma_{\perp,k} \mathrm{V}_{\perp,k}^\top \mathrm{V}) \\
&\geq \mathbf{tr}(\mathrm{V}^\top \mathrm{V}_{\perp,k} \Gamma_{\perp,k} \mathrm{V}_{\perp,k}^\top \mathrm{V}) \\
&= \mathbf{tr}(\mathrm{V}_{\perp,k}^\top \mathrm{V} \mathrm{V}^\top \mathrm{V}_{\perp,k} \cdot \Gamma_{\perp,k}) \geq A \cdot \||\mathrm{V}_{\perp,k}^\top \mathrm{V}\||_F^2.
\end{aligned}
$$

From this, it follows that $F(\mathrm{V}) \geq \max_k A\||\mathrm{V}_{\perp,k}^\top \mathrm{V}\||_F^2$.

Now, let $L = \max_k d_F(\mathrm{V}, \mathrm{V}_{J,k})$. By Proposition 4, it holds that $\sqrt{2}\||\mathrm{V}^\top[\mathrm{V}_{I,k}, \mathrm{V}_{\perp,k}]\||_F = d_F(\mathrm{V}, \mathrm{V}_{J,k})$. Therefore, there exists an index $k_0$

such that $\sqrt{2}\||\mathrm{V}^\top[\mathrm{V}_{I,k_0}, \mathrm{V}_{\perp,k_0}]\||_F = L$. From this, $\||\mathrm{V}^\top \mathrm{V}_{I,k_0}\||_F^2 + \||\mathrm{V}^\top \mathrm{V}_{\perp,k_0}\||_F^2 = \frac{L^2}{2}$. Let us show that $\max_k \||\mathrm{V}_{\perp,k}^\top \mathrm{V}\||_F^2 \geq \frac{L^2}{6}$.

If $\||\mathrm{V}_{\perp,k_0}^\top \mathrm{V}\||_F^2 > \frac{L^2}{6}$, then we conclude. Otherwise, we assume that $\||\mathrm{V}_{\perp,k_0}^\top \mathrm{V}\||_F^2 \leq \frac{L^2}{6}$. Then, $\||\mathrm{V}_{I,k_0}^\top \mathrm{V}\||_F^2 \geq \frac{L^2}{3}$. By Assumption 6, there exists $\ell \neq k_0$ such that $\||\mathrm{V}_{I,k_0}^\top \mathrm{V}_{H,\ell}\||_F \leq \varepsilon_{2,p}$. By Proposition 5, there exists $\mathrm{V}_{k_0,\ell}^* \in \mathbb{R}^{p\times s_k}$ an $s_k$-frame such that $\||\mathrm{V}_{I,k_0} - \mathrm{V}_{k_0,\ell}^*\||_F \leq 2\||\mathrm{V}_{I,k_0}^\top \mathrm{V}_{H,\ell}\||_F \leq 2\varepsilon_{2,p}$ and $\mathbf{span}(\mathrm{V}_{k_0,\ell}^*) \subset \mathbf{span}(\mathrm{V}_{H,\ell})^\perp = \mathbf{span}(\mathrm{V}_{\perp,\ell})$. Combined with the assumption that $L > 12\varepsilon_{2,p}$,

$$
\begin{aligned}
\||\mathrm{V}^\top \mathrm{V}_{\perp,\ell}\||_F &\geq \||\mathrm{V}^\top \mathrm{V}_{k_0,\ell}^*\||_F \\
&\geq \||\mathrm{V}^\top \mathrm{V}_{I,k_0}\||_F - \||\mathrm{V}^\top(\mathrm{V}_{k_0,\ell}^* - \mathrm{V}_{I,k_0})\||_F \\
&\geq \frac{L}{\sqrt{3}} - 2\varepsilon_{2,p} \geq \frac{L}{\sqrt{3}} - \frac{L}{6} \geq \frac{L}{\sqrt{6}}.
\end{aligned}
$$

From this, $\max_k \||\mathrm{V}^\top \mathrm{V}_{\perp,k}\||_F^2 \geq \frac{L^2}{6}$. To conclude the proof, we simply observe that $\max_k \||\Sigma^{(k)}\||_2 \leq CA$, and therefore, for $c_2^* = 12$ and $c_3^* = 1/6C$,

$$
\begin{aligned}
F(\mathrm{V}) &\geq \max_k A\||\mathrm{V}_{\perp,k}^\top \mathrm{V}\||_F^2 \geq \frac{AL^2}{6} \\
&\geq c_3^* \max_k \||\Sigma^{(k)}\||_2 \cdot \max_k d_F^2(\mathrm{V}, \mathrm{V}_{J,k}).
\end{aligned}
$$

$\qquad\square$

In Proposition 2 we show the quadratic growth of the objective function with respect to the maximum distance $\max_k d_F(\mathrm{V}, \mathrm{V}_{J,k})$. Our goal is to have a similar quadratic bound that depends on the distance $d_F(\mathrm{V}, \mathrm{V}_s^\#)$. To achieve this, we first need to establish the distance of the $s$-frame $\mathrm{V}_s^\#$ to the joint hub frames $\{\mathrm{V}_{J,k}\}_{k\in[K]}$. This is covered by the following lemma.

**Lemma 3.** *Let $\{\Theta^{(k)}\}_k$ be matrices satisfying Assumptions 1, 2 and 6, and $\mathrm{V}_s^\#$ be the minimizer, as defined in (2). Then, there exists constants $c_4^* > 0$ such that,*

$$
\max_k d_F(\mathrm{V}_s^\#, \mathrm{V}_{J,k}) \leq c_4^* \cdot \max\left\{\varepsilon_{2,p}, p^{-\beta/4}\right\}.
$$

*Proof.* By Lemma 1, we know that there exists a constant $c_1^* > 0$ such that, for a sufficiently large $p$, $F(\mathrm{V}_s^\#) \leq c_1^* \cdot a$. On the other hand, by Lemma 2, we know that if an $s$-frame $\mathrm{V}$ satisfies $\max_k d_F(\mathrm{V}, \mathrm{V}_{J,k}) \geq c_2^* \varepsilon_{2,p}$, then $F(\mathrm{V}) \geq c_3^* \max_k \||\Sigma^{(k)}\||_2 \cdot \max_k d_F^2(\mathrm{V}, \mathrm{V}_{J,k})$. Combining these results, we find that if $\max_k d_F(\mathrm{V}, \mathrm{V}_{J,k}) > \max\{c_2^* \varepsilon_{2,p}, \sqrt{c_1^* a/c_3^* \max_k \||\Sigma^{(k)}\||_2}\}$, then $F(\mathrm{V}) >$

$c_1^* a$. By Assumption 2, we have that $a / \max_k \left\| \left\| \Sigma^{(k)} \right\| \right\|_2 \leq a/A \leq c^* p^{-\beta/2}$, for some $c^* > 0$. Therefore, there exists a constant $c_4^* > 0$ such that $\max_k d_F(V_s^\#, V_{J,k}) \leq c_4^* \cdot \max\{\varepsilon_{2,p}, p^{-\beta/4}\}$. $\square$

By combining the uncentered quadratic growth shown in Lemma 2 and the bound on the distance shown in Lemma 3, we can show quadratic growth of our estimator as a function of the distance to $V_s^\#$.

**Lemma 4.** *Let $\{\Theta^{(k)}\}_k$ be matrices satisfying Assumptions 1, 2 and 6, and $V_s^\#$ be the minimizer, as defined in (2). Then, there exist $c_5^* > 0$ and $c_6^* > 2s$ such that, for all s-frame $V \in \mathbb{R}^{p \times s}$ that satisfies $d_F(V, V_s^\#) \geq 2c_5^* \max\left\{\varepsilon_{2,p}, p^{-\beta/4}\right\}$, then,*

$$F(V) - F(V_s^\#) \geq c_6^* \max_k \left\| \left\| \Sigma^{(k)} \right\| \right\|_2 d_F^2(V, V_s^\#).$$

*Proof.* Assume $d_F(V, V_s^\#) \geq 2c_4^* \max\left\{\varepsilon_{2,p}, p^{-\beta/4}\right\}$. By applying Lemma 3,

$$\max_k d_F(V, V_{J,k}) \geq \max_k \left\{ d_F(V, V_s^\#) - d_F(V_s^\#, V_{J,k}) \right\}$$
$$\geq c_4^* \max\left\{\varepsilon_{2,p}, p^{-\beta/4}\right\} \geq c_4^* \varepsilon_{2,p}.$$

Furthermore, observe that,

$$d_F(V, V_{J,k}) \geq d_F(V, V_s^\#) - d_F(V_s^\#, V_{J,k})$$
$$\geq d_F(V, V_s^\#) - c_4^* \max\left\{\varepsilon_{2,p}, p^{-\beta/4}\right\}$$
$$\geq \frac{d_F(V, V_s^\#)}{2}.$$

From this, $V$ satisfies the assumptions of Lemma 2. Therefore, by Lemmas 1 and 2,

$$F(V) - F(V_s^\#)$$
$$\geq c_3^* \max_k \left\| \left\| \Sigma^{(k)} \right\| \right\|_2 \max_k d_F^2(V, V_{J,k}) - c_1^* a$$
$$\geq \frac{c_3^* \max_k \left\| \left\| \Sigma^{(k)} \right\| \right\|_2 \cdot d_F^2(V, V_s^\#)}{4} - c_1^* a.$$

Since $d_F^2(V, V_s^\#) \geq 4(c_4^*)^2 p^{-\beta/2}$, and $a / \max_k \left\| \left\| \Sigma^{(k)} \right\| \right\|_2 \leq a/A \leq c^* p^{-\beta/2}$, for some $c^* > 0$. Then, as long as we select a new value $c_5^* > c_4^*$ such that $2c_5^* > 4c_1^* c^*/c_3^*$, then

$$d_F^2(V, V_s^\#) - \frac{4c_1^* a}{c_3^* \max_k \left\| \left\| \Sigma^{(k)} \right\| \right\|_2} > \frac{d_F^2(V, V_s^\#)}{2}.$$

Thus, for sufficiently large $p$, $F(V) - F(V_s^\#) \geq c_6^* \cdot \max_k \left\| \left\| \Sigma^{(k)} \right\| \right\|_2 \cdot d_F^2(V, V_s^\#)$ for an appropriate choice of $c_6^* > 0$. In particular, we select $c_6^* > 2s$. $\square$

### B.3. Convergence Rate for the Objective Function

**Lemma 5.** *Let $M_k[V] = \widehat{F}_k(V) - F_k(V)$. Then, for any $R > 0$,*

$$\sup_{V \in \mathrm{St}(p,s)} |M_k[V]| \leq s \cdot \mathcal{E}(n_k, p, t) \cdot \left\| \left\| \Sigma^{(k)} \right\| \right\|_2$$

*with probability at least $1 - c_2 e^{-c_3 n \min\{t, t^2\}}$*

*Proof.* Observe that, for any $V^\top V = I_s$,

$$|M_k[V]| = |\widehat{F}_k(V) - F_k(V)|$$
$$= \left| \mathbf{tr}\left( VV^\top (\widehat{\Sigma}^{(k)} - \Sigma^{(k)}) \right) \right|$$
$$\leq s \cdot \left\| \left\| \widehat{\Sigma}^{(k)} - \Sigma^{(k)} \right\| \right\|_2.$$

By Assumption 4, we have that $\mathbb{P}\left[ \left\| \left\| \widehat{\Sigma}^{(k)} - \Sigma^{(k)} \right\| \right\|_2 \geq \mathcal{E}(n_k, p, t) \cdot \left\| \left\| \Sigma^{(k)} \right\| \right\|_2 \right] \leq c_2 e^{-c_3 n \min\{t, t^2\}}$. From this,

$$\mathbb{P}\left[ \sup_{V \in \mathrm{St}(p,s)} |M_k[V]| \geq s \cdot \mathcal{E}(n_k, p, t) \cdot \left\| \left\| \Sigma^{(k)} \right\| \right\|_2 \right]$$
$$\leq \mathbb{P}\left[ \left\| \left\| \widehat{\Sigma}^{(k)} - \Sigma^{(k)} \right\| \right\|_2 \geq \mathcal{E}(n_k, p, t) \cdot \left\| \left\| \Sigma^{(k)} \right\| \right\|_2 \right]$$
$$\leq c_2 e^{-c_3 n \min\{t, t^2\}}.$$

$\square$

We can combine the individual rates of convergence, as well as our previous useful tools, to obtain a rate of convergence of the joint process.

**Lemma 6.** *For any $t > 0$,*

$$\sup_{V \in \mathrm{St}(p,s)} | \max_{k \in [K]} \widehat{F}_k(V) - \max_{k \in [K]} F_k(V)|$$
$$\leq s \cdot \mathcal{E}(n, p, t) \cdot \max_k \left\| \left\| \Sigma^{(k)} \right\| \right\|_2$$

*with probability at least $1 - Kc_2 e^{-c_3 n \min\{t, t^2\}}$.*

*Proof.* First, by denoting $\mathbb{K} := \max_k \left\| \left\| \widehat{\Sigma}^{(k)} \right\| \right\|_2$, observe that, when taking the supremum over $V \in \mathrm{St}(p,s)$,

$$\mathbb{P}\left[ \sup_V | \max_k \widehat{F}_k(V) - \max_k F_k(V)| \geq s\mathcal{E}(n, p, t)\mathbb{K} \right]$$
$$\leq \mathbb{P}\left[ \max_k \sup_V |\widehat{F}_k(V) - F_k(V)| \geq s\mathcal{E}(n, p, t)\mathbb{K} \right]$$
$$\leq \sum_{k=1}^K \mathbb{P}\left[ \sup_V |M_k(V)| \geq s\mathcal{E}(n, p, t) \left\| \left\| \Sigma^{(k)} \right\| \right\|_2 \right].$$

For each $k \in [K]$,

$$\sup_{V \in \mathrm{St}(p,s)} |M_k(V)| \geq s\mathcal{E}(n,p,t)\left\|\left\|\Sigma^{(k)}\right\|\right\|_2$$

with probability at most $c_2 e^{-c_3 n \min\{t, t^2\}}$. From this, we conclude. $\square$

### B.4. Joint Eigenvector Consistency Proof

The following is the central result in our theoretical development. The guarantees in the convergence of eigenvectors, as well as the guarantees in the rate of detection of common hubs is derived from this result. The proof is based on the proof of consistency of M-estimators found in Chapter 5 of Sen (2018).

**Theorem.** Assume the family of precision matrices $\{\Theta^{(k)}\}_{k \in [K]}$ satisfies Assumptions 1, 2 and 3, and our covariance matrix estimators $\{\widehat{\Sigma}^{(k)}\}_{k \in [K]}$ satisfy Assumption 4. Then there exists a constant $c_4 > 0$ such that for any $t > 0$, our estimator $\widehat{V}_s^\# \in \mathbb{R}^{p \times s}$ based on the true value of $s$ satisfies,

$$d_F(\widehat{V}_s^\#, V_s^\#) \leq c_4 \max\left\{\varepsilon_p, p^{-\beta/4}, \sqrt{c_1 \mathcal{E}(n,p)}\right\} + t,$$

with probability at least $1 - Kc_2 e^{-c_3 n \min\{t^2/c_1, t^4/c_1^2\}}$.

*Proof of Theorem 1.* By Proposition 1, our matrices satisfy Assumption 6 for the sequence $\varepsilon_{2,p} = C \max\{\varepsilon_p, p^{-\beta/4}\}$. Let $R = c_5^* \max\left\{\varepsilon_{2,p}, p^{-\beta/4}, \sqrt{c_1 \mathcal{E}(n,p)}\right\}$. Notice that $R \leq c_5^{**} \max\{\varepsilon_p, p^{-\beta/4}, \sqrt{c_1 \mathcal{E}(n,p)}\}$ for some constant $c_5^{**} > 0$. Now, since the bound $R \geq c_5^* \max\left\{\varepsilon_{2,p}, p^{-\beta/4}\right\}$ holds, by Lemma 4, for any $V \in \mathrm{St}(p,s)$ with $d_F(V, V_s^\#) > R$,

$$F(V) - F(V_s^\#) > c_6^* \max_k \left\|\left\|\widehat{\Sigma}^{(k)}\right\|\right\|_2 \cdot d_F^2(V, V_s^\#).$$

Now, we denote $\mathbb{F} = \sup_{V \in \mathrm{St}(p,s)} |\widehat{F}(V) - F(V)|$, where $F(V) = \max_{k \in [K]}\{\mathbf{tr}(V^\top \Sigma^{(k)} V)\}$ and $\widehat{F}(V) = \max_{k \in [K]}\{\mathbf{tr}(V^\top \widehat{\Sigma}^{(k)} V)\}$.

Let $t > 0$. Then, we observe that if $d_F(\widehat{V}_s^\#, V_s^\#) > R + t$ holds,

$$\begin{aligned}
\widehat{F}(\widehat{V}_S^\#) &\geq F(\widehat{V}_S^\#) - \mathbb{F} \\
&\geq F(V_S^\#) + c_6^* \max_k \left\|\left\|\widehat{\Sigma}^{(k)}\right\|\right\|_2 \cdot d_F^2(V, V_s^\#) - \mathbb{F} \\
&\geq \widehat{F}(V_s^\#) + c_6^* \max_k \left\|\left\|\widehat{\Sigma}^{(k)}\right\|\right\|_2 \cdot d_F^2(\widehat{V}_s^\#, V_s^\#) - 2\mathbb{F} \\
&\geq \widehat{F}(V_s^\#) + c_6^* \max_k \left\|\left\|\widehat{\Sigma}^{(k)}\right\|\right\|_2 \cdot (R+t)^2 - 2\mathbb{F}.
\end{aligned}$$

Since $\widehat{V}_s^\#$ is the minimizer of $\widehat{F}$,

$$\begin{aligned}
0 &> \widehat{F}(\widehat{V}_s^\#) - \widehat{F}(V_s^\#) \\
&\geq c_6^* \max_k \left\|\left\|\widehat{\Sigma}^{(k)}\right\|\right\|_2 \cdot (R+t)^2 - 2\mathbb{F}.
\end{aligned}$$

Therefore, if we denote $\mathbb{K} := \max_k \left\|\left\|\widehat{\Sigma}^{(k)}\right\|\right\|_2$

$$\begin{aligned}
&P(d_F(\widehat{V}_s^\#, V_s^\#) > R) \\
&\leq P\left(\sup_{V \in \mathrm{St}(p,s)} |\widehat{F}(V) - F(V)| > \frac{\mathbb{K}c_6^*(R+t)^2}{2}\right).
\end{aligned}$$

Now, since $R \geq \sqrt{c_1 \mathcal{E}(n,p)}$, we have that $(R+t)^2 \geq R^2 + t^2 = c_1(\mathcal{E}(n,p) + t^2/c_1) = \mathcal{E}(n,p,t^2/c_1)$. Finally, since $c_6^* > 2s$, and by Lemma 6,

$$\begin{aligned}
&P(d_F(\widehat{V}_s^\#, V_s^\#) > R) \\
&\leq P\left(\sup_{V \in \mathrm{St}(p,s)} |\widehat{F}(V) - F(V)| > \frac{\mathbb{K}c_6^*(R+t)^2}{2}\right) \\
&\leq P\left(\sup_{V \in \mathrm{St}(p,s)} |\widehat{F}(V) - F(V)| > \mathbb{K}s\mathcal{E}(n,p,t^2/c_1)\right) \\
&\leq c_2 e^{c_3 n \min\{t^2/c_1, t^4/c_1^2\}}
\end{aligned}$$

From this, we conclude. $\square$

### B.5. Common Hub Estimation Guarantees

**Lemma 7.** *Consider the setups and assumptions as in Theorem 1. Further assume that $\beta > 1/2$, $V_{J,k} = V_J$ for all $k \in [K]$, and the minimax eigenspace dimension $s$ is correctly specified as the common eigenspace dimension. Then, there exists $c_7^* > 0$ such that, for a sufficiently large $p$,*

$$\max_{k \notin \mathcal{J}} \widehat{\omega}_s^\#(\ell) \leq c_7^* \max\left\{\varepsilon_p, p^{-\beta/4}, \sqrt{c_1 \mathcal{E}(n,p)}\right\} + t.$$

*with probability at least $1 - Kc_2 e^{-c_3 n \min\{t^2/c_1, t^4/c_1^2\}}$.*

*Proof.* For each $k \in [K]$, let $V_{H,k} \in \mathbb{R}^{p \times S_k}$ be the matrix of leading $S_k$-eigenvalues of $\Theta^{(k)}$. Furthermore, we denote the individual influence measures of $\Theta^{(k)}$ by $\{\omega_{S_k}^{(k)}(\ell) := [V_{H,k}^\top V_{H,k}]_{\ell\ell}\}_{\ell \in [p]}$.

Let $k \in [K]$ fixed. Since $\{\Theta^{(k)}\}_{k \in [K]}$ satisfies Assumption 1 for $\beta > 1/2$, and Assumption 2 each of the precision matrices $\Theta^{(k)} \in \mathbb{R}^{p \times p}$ satisfy the assumptions of Theorem 1 from Sánchez Gómez et al. (2025). From this, we have that $\max_{\ell \notin \mathcal{H}^k} \omega_{S_k}^{(k)}(\ell) = O(p^{-1})$, where $\mathcal{H}^k = \mathcal{J} \cup \mathcal{I}^k$. Recall that $\omega_{S_k}^{(k)}(\ell) = [V_{H,k} V_{H,k}^\top]_{\ell\ell}$, and $\omega_s^\#(\ell) = [V_s^\#(V_s^\#)^\top]_{\ell\ell}$. Furthermore, we denote $\omega^{(J,k)}(\ell) = [V_{J,k} V_{J,k}^\top]_{\ell\ell}$ and $\omega_k^{(I)}(\ell) = [V_{I,k} V_{I,k}^\top]_{\ell\ell}$ for each $k \in [K]$ and $\ell \in \mathcal{P}$.

Since $\mathbf{span}(\mathrm{V}_{H,k}) = \mathbf{span}(\mathrm{V}_{I,k}) \oplus \mathbf{span}(\mathrm{V}_{J,k})$, then $\omega_{S_k}^{(k)}(\ell) = \omega^{(J,k)}(\ell) + \omega^{(I,k)}(\ell)$ for any $\ell \in \mathcal{P}$. Notice that, for any $\ell \notin \mathcal{J}$, there exists a population $k_*$ such that $\ell$ is not a hub for $\Theta^{(k_*)}$, so $\ell \notin \tilde{\mathcal{I}}^{k_*}$. From this, it follows directly that $\omega^{(J,k_*)}(\ell) \leq \omega_{S_{k_*}}^{(k_*)} = O(p^{-1})$. Therefore, $\max_{\ell \notin \mathcal{J}} \min_{k \in [K]} \omega^{(J,k)}(\ell) = O(p^{-1})$. By applying Lemma 3, there exists a constant $c^* > 0$ such that

$$\max_{\ell \notin \mathcal{J}} \omega_s^\#(\ell) = \max_{\ell \notin \mathcal{J}} [\mathrm{V}_s^\# (\mathrm{V}_s^\#)^\top]_{\ell\ell}$$
$$\leq \max_{\ell \notin \mathcal{J}} \min_{k \in [K]} \left\{ \| \mathrm{V}_s^\# (\mathrm{V}_s^\#)^\top - \mathrm{V}_{J,k} \mathrm{V}_{J,k}^\top \|_{\max} + \right.$$
$$\left. [\mathrm{V}_{J,k} \mathrm{V}_{J,k}^\top]_{\ell\ell} \right\}$$
$$\leq \max_{k \in [K]} d_F(\mathrm{V}_s^\#, \mathrm{V}_J) + O(p^{-1})$$
$$= c^* \max \left\{ \varepsilon_{2,p}, p^{-\beta/4} \right\}.$$

To conclude, recall $\widehat{\omega}_s^\#(\ell) = [\widehat{\mathrm{V}}_s^\# (\widehat{\mathrm{V}}_s^\#)^\top]_{\ell\ell}$ for $\ell \in \mathcal{P}$. Therefore, applying Theorem 1, and Proposition 1, for any $t > 0$,

$$\max_{\ell \notin \mathcal{J}} \widehat{\omega}_s^\#(\ell) = \max_{\ell \notin \mathcal{J}} [\widehat{\mathrm{V}}_s^\# (\widehat{\mathrm{V}}_s^\#)^\top]_{\ell\ell}$$
$$\leq \| \widehat{\mathrm{V}}_s^\# (\widehat{\mathrm{V}}_s^\#)^\top - \mathrm{V}_s^\# (\mathrm{V}_s^\#)^\top \|_{\max} + \max_{\ell \notin \mathcal{J}} \omega_s^\#(\ell)$$
$$\leq (c^* + c_4) \max \left\{ p^{-\beta/4}, \sqrt{c_1 \mathcal{E}(n,p)} \right\} + t$$
$$\leq c_7 \max \left\{ \varepsilon_p, p^{-\beta/4}, \sqrt{c_1 \mathcal{E}(n,p)} \right\} + t$$

with probability at least $1 - Kc_2 \exp\left\{ -c_3 n \min\{t^2/c_1, t^4/c_1^2\} \right\}$. $\square$

**Theorem.** Consider the setups and assumptions as in Theorem 1. Further assume that $\beta > 1/2$, and the minimax eigenspace dimension $s$ is correctly specified as the common eigenspace dimension. Then there exists a universal constant $c_5 > 0$, such that for any influence measure threshold $\kappa \geq c_5 \max\left\{ \varepsilon_p, p^{-\beta/4}, \sqrt{c_1 \mathcal{E}(n,p)} \right\}$, we have

$$\mathbb{P}\left\{ \widehat{\mathcal{J}}(\kappa) \subseteq \mathcal{J} \right\} \geq 1 - Kc_2 \cdot e^{-c_3 n \min\{\mathcal{E}(n,p), \mathcal{E}^2(n,p)\}}.$$

*Proof.* This result follows from applying Lemma 7 with $t = c_7 \max\left\{ \varepsilon_p, p^{-\beta/4}, \sqrt{c_1 \mathcal{E}(n,p)} \right\}$. $\square$

## B.6. Sufficient Conditions for Assumption 3 and Perfect Recovery

First, we show that if the family of matrices $\{\Theta^{(k)}\}_{k=1}^K$ have bounded differences, our eigenspace decomposability Assumption 3 is satisfied. For this, we provide the following condition.

**Assumption 7** (Pairwise Bounded Differences). *Assume there exists a positive definite matrix $\Theta^{(\mathcal{J})} \in \mathbb{R}^{p \times p}$ such*

*that the family of matrices $\{\Theta^{(1)}, \Theta^{(2)}, \ldots, \Theta^{(K)}, \Theta^{(\mathcal{J})}\}$ satisfy Assumptions 1 and 2. Furthermore, the matrices $\Theta^{(Rk)} := \Theta^{(k)} - \Theta^{(\mathcal{J})}$ satisfy $\max_{k \in [K]} \left\| \left| \Theta^{(Rk)} \right| \right\|_2 = O(1/A_p).$*

**Proposition 6.** *Let $\{\Theta^{(k)}\}_{k \in [K]}$ precision matrices that satisfy Assumptions 1, 2. Furthermore, Assumption 7 holds. Then, the matrices $\{\Theta^{(k)}\}_{k \in [K]}$ satisfy Assumption 3.*

*Proof.* For any symmetric matrix $\mathrm{A} \in \mathbb{R}^{p \times p}$, we denote by $\lambda_1(\mathrm{A}) \geq \lambda_2(\mathrm{A}) \geq \ldots \geq \lambda_p(\mathrm{A})$ the ordered eigenvalues of $\mathrm{A}$. By Assumption 2, we have that $\lambda_1(\Theta^{(\mathcal{J})}), \ldots, \lambda_s(\lambda_1(\Theta^{(\mathcal{J})})) \sim 1/a_p$ and $\lambda_{s+1}(\Theta^{(\mathcal{J})}), \ldots, \lambda_p(\lambda_1(\Theta^{(\mathcal{J})})) \sim 1/A_p$ with $A_p/a_p \sim p^{\beta/2}$. By Assumption 7 and Weyl's inequality, we have that for any $k \in [K]$ and $i \in [s]$, $\lambda_i(\Theta^{(k)}) = \lambda_i(\Theta^{(\mathcal{J})}) + O(1/A_p) \sim 1/a_p$. Similarly, we have that for $s+1 \leq i \leq p$, $\lambda_i(\Theta^{(k)}) = \lambda_i(\Theta^{(\mathcal{J})}) + O(1/A_p) = O(1/A_p)$. From this, it follows that all matrices $\{\Theta^{(k)}\}_{k=1}^K$ have the same number of separating eigenvalues $s$, i.e. $S_1 = S_2 = \ldots = S_K = s$.

Since $\lambda_{s+1}(\Theta^{(\mathcal{J})}) = o(\lambda_s(\Theta^{(\mathcal{J})}))$, we have that $\lambda_s(\Theta^{(\mathcal{J})}) - \lambda_{s+1}(\Theta^{(\mathcal{J})}) \sim 1/a_p$. For each $k \in [K]$, let $\mathrm{V}_{Jk} \in \mathbb{R}^{p \times s}$ be the matrix with the leading $s$ eigenvectors of $\Theta^{(k)}$. Similarly, we denote by $\mathrm{V}_{\mathcal{J}} \in \mathbb{R}^{p \times s}$ be the matrix with the leading $s$ eigenvectors of $\Theta^{(\mathcal{J})}$. Then, by Davis-Kahan's Theorem (Fan et al. (2021), Corollary 2.1) and since $\left\| \left| \Theta^{(Rk)} \right| \right\|_2 = o(1/A_p)$,

$$d_F(\mathrm{V}_{Jk}, \mathrm{V}_{\mathcal{J}}) \leq \sqrt{s} \left\| \left| \mathrm{V}_{Jk} \mathrm{V}_{Jk}^\top - \mathrm{V}_{\mathcal{J}} \mathrm{V}_{\mathcal{J}}^\top \right| \right\|_2$$
$$\leq \frac{\sqrt{4s} \left\| \left| \Theta^{(Rk)} \right| \right\|_2}{\lambda_s(\Theta^{(\mathcal{J})}) - \lambda_{s+1}(\Theta^{(\mathcal{J})})}$$
$$= O\left( \frac{a_p}{A_p} \right) = O(p^{-\beta/2}).$$

Thus, we conclude that $\max_{k \neq \ell} d_F(\mathrm{V}_{Jk}^\top, \mathrm{V}_{J\ell}) = O(p^{-\beta/2}) = o(p^{-\beta/4})$. Thus, condition A3-1 holds. By setting $\mathrm{V}_{I1} = \mathrm{V}_{I2} = \ldots = \mathrm{V}_{IK} = \mathbf{0}$, condition A3-2 is also satisfied. Thus, we conclude Assumption 3 is satisfied with $\varepsilon_p \equiv 0$. $\square$

**Theorem 3.** *Assume that the precision matrices $\{\Theta^{(k)}\}_{k \in [K]}$ for multiple GGMs satisfy Assumptions 1, 2, 7, the covariance estimators $\{\widehat{\Sigma}^{(k)}\}_{k \in [K]}$ satisfy Assumption 4 and the minimax eigenspace dimension $s$ is correctly specified as the common eigenspace dimension. Then, if $\beta > 4/5$ and $\mathcal{E}(n,p) = O(p^{-\beta/2})$, there exists a universal constants $c_5, c_6 > 0$, such that for $t > 0$ such that $\kappa := c_5 p^{-\beta/4} + t \leq c_6 p^{1-\beta} - t$, we have*

$$\mathbb{P}\left\{ \widehat{\mathcal{J}}(\kappa) = \mathcal{J} \right\} \geq 1 - Kc_2 \cdot e^{-c_3 n \min\{t^2/c_1, t^4/c_1^2\}}.$$

*Proof.* To simplify our notation, we consider scaling constants to adjust according to the steps of the proof.

By Proposition 6, the matrices $\{\Theta^{(k)}\}_{k\in[K]}$ satisfy all conditions of Theorem 1 with $\varepsilon_p \equiv 0$. From this, for any $t > 0$, with probability at least $1 - Kc_2 \exp\{-c_3 n \min\{t^2/c_1, t^4/c_1^2\}\}$,

$$d_F(\widehat{\mathrm{V}}_s^{\#}, \mathrm{V}_s^{\#}) \leq c_4 \max\left\{p^{-\beta/4}, \sqrt{c_1 \mathcal{E}(n,p)}\right\} + t$$
$$\leq c_5 \cdot p^{-\beta/4} + t,$$

since $\mathcal{E}(n,p) = O(p^{-\beta/2})$. Furthermore, by the proof of Proposition 6, we have that $\max_{k\in[K]} d_F(\mathrm{V}_{\mathcal{J}}, \mathrm{V}_{Jk}) = O(p^{-\beta/2})$. By Lemma 3, we have that $\max_k d_F(\mathrm{V}_s^{\#}, \mathrm{V}_{\mathcal{J}}) = O(p^{-\beta/4})$. From this, it follows that with high probability, $d_F(\widehat{\mathrm{V}}_s^{\#}, \mathrm{V}_{\mathcal{J}}) \leq c_5 \cdot p^{-\beta/4} + t$.

For the matrix $\Theta^{(\mathcal{J})}$, we define the influence measures $\omega^{(\mathcal{J})}(\ell) = [\mathrm{V}_{\mathcal{J}}\mathrm{V}_{\mathcal{J}}^{\top}]_{\ell\ell}$. Since $\Theta^{(\mathcal{J})}$ contains the set of hubs $\mathcal{J}$ and satisfies $\lambda_{s+1}(\Theta^{(\mathcal{J})})/\lambda_p(\Theta^{(\mathcal{J})}) = O(1)$ and $\beta > 1/2$, by Theorem 1 of Sánchez Gómez et al. (2025), we have that, $\max_{\ell\notin\mathcal{J}} \omega^{(\mathcal{J})}(\ell) = O(p^{-1})$. Since $p^{-1} \ll p^{-\beta/4}$, it follows that with high probability,

$$\max_{\ell\notin\mathcal{J}} \widehat{\omega}_s^{\#}(\ell) \leq \max_{\ell\notin\mathcal{J}}\{\omega^{(\mathcal{J})}(\ell)\} + d_F(\widehat{\mathrm{V}}_s^{\#}, \mathrm{V}_{\mathcal{J}})$$
$$\leq c_5 p^{-\beta/4} + t.$$

From this, if $\kappa = c_5 p^{-\beta/4} + t$, we have that $\widehat{\mathcal{J}}(\kappa) \subset \mathcal{J}$ with a probability of at least $1 - Kc_2 \exp\{-c_3 n \min\{t^2/c_1, t^4/c_1^2\}\}$.

Similarly, by Theorem 1 of Sánchez Gómez et al. (2025), we have that, $\min_{h\in\mathcal{J}} \omega^{(\mathcal{J})}(h) = \Omega(p^{1-\beta})$. Since $\beta > 4/5$, it follows that $p^{1-\beta} \gg p^{-\beta/4}$. Therefore, for some constant $c_6 > 0$,

$$\min_{h\in\mathcal{J}} \widehat{\omega}_s^{\#}(h) \geq \min_{h\in\mathcal{J}} \omega^{(\mathcal{J})}(h) - d_F(\widehat{\mathrm{V}}_s^{\#}, \mathrm{V}^{(\mathcal{J})})$$
$$\geq \min_{h\in\mathcal{J}} \omega^{(\mathcal{J})}(h) - c_5 p^{-\beta/4} - t$$
$$\geq c_6 p^{1-\beta} - t,$$

with high probability. As a consequence, for any $t > 0$ such that $c_5 p^{-\beta/4} + 2t < c_6 p^{1-\beta}$, we have that for $\kappa = c_5 p^{-\beta/4} + t$,

$$\mathbb{P}\{\widehat{\mathcal{J}}(\kappa) = \mathcal{J}\} \geq 1 - Kc_2 \cdot e^{-c_3 n \min\{t^2/c_1, t^4/c_1^2\}}.$$

$\square$

## C. Estimation of Joint Eigenspace Dimension $s$

In high-dimensional scenarios, where $n_1, n_2, \ldots, n_K \ll p$, the estimation of the joint eigenspace dimension $s$ may be challenging. As shown in simulations, the use of an overestimation $\hat{s} := \lfloor\sqrt{p}\rfloor$ can reliably recover common hubs

across multiple populations. Similar use of the overestimation of the dimension of common eigenspaces have been applied in the context of distributed PCA (Chen et al., 2022). This method is applied directly in our systematic numerical simulation studies found in Section 5.

For sufficiently large sample sizes, we may estimate $s$, with the method described below. Our procedure for the estimation of $s$ is based on the following motivation. Consider $\Theta^{(1)}, \ldots, \Theta^{(K)} \in \mathbb{R}^{p\times p}$ precision matrices with associated covariance matrices $\{\Sigma^{(k)}\}_{k\in[K]}$, with a set of common hubs captured by a matrix $\Theta_J \in \mathbb{R}^{p\times p}$ with spectral decomposition $\Theta_J = \sum_{i=1}^{s} \lambda_i \boldsymbol{v}_i \boldsymbol{v}_i^{\top}$, with $\lambda_1, \lambda_2, \ldots, \lambda_s = \Omega(1/a_p)$ as in Assumption 2. It can be shown that if $\mathrm{V} = [\boldsymbol{v}_1, \boldsymbol{v}_2, \ldots \boldsymbol{v}_s]$, $F(\boldsymbol{v}_i) = \mathrm{argmin}_{k\in[K]} \{\mathbf{tr}(\boldsymbol{v}_i^{\top}\Sigma^{(k)}\boldsymbol{v}_i)\} \approx 1/\lambda_i = O(a_p)$, and $F(\mathrm{V}) = \mathrm{argmin}_{k\in[K]} \{\mathbf{tr}(\mathrm{V}^{\top}\Sigma^{(k)}\mathrm{V})\} \approx \sum_{i=1}^{s} 1/\lambda_i = O(a_p)$. Furthermore, if $\boldsymbol{v} \perp \mathrm{V}$, we have that $F(\boldsymbol{v}) \approx A_p \gg a_p$. Thus, we expect a large gap between the objective function value for joint eigenspaces, and any remaining orthogonal subspace.

For sufficiently large sample sizes, it may be possible to accurately estimate the value of $\hat{s}$. The procedure is as follows. First, we calculate the minimizers $\{\widehat{\mathrm{V}}_i^{\#} : 1 \leq i \leq \lfloor\sqrt{p}\rfloor\}$ and the values $\left\{\hat{\eta}_i := \mathrm{argmin}_{k\in[K]} \mathbf{tr}((\widehat{\mathrm{V}}_i^{\#})^{\top}\Sigma^{(k)}\widehat{\mathrm{V}}_i^{\#}) : 1 \leq i \leq \lfloor\sqrt{p}\rfloor\right\}$. Then, we define the estimated joint minimax eigenvalues as $\hat{\lambda}_1^{\#} = 1/\hat{\eta}_1$, and $\hat{\lambda}_i^{\#} = 1/(\hat{\eta}_i - \hat{\eta}_{i-1})$ for any $2 \leq i \leq \lfloor\sqrt{p}\rfloor$. Furthermore, we define the consecutive eigenvalue ratio gaps as $\delta_i := \hat{\lambda}_i^{\#}/\hat{\lambda}_{i+1}^{\#}$. To estimate $\hat{s}$, we set $\hat{s}_{pre} = \mathrm{argmax}_{1\leq i\leq\lfloor\sqrt{p}\rfloor-1} \delta_i$. Then, we set $\hat{s} = \hat{s}_{pre}\mathbb{1}(\hat{s}_{pre} > 1.5\max_{i\neq\hat{s}_{pre}} \delta_i) + \lfloor p\rfloor \cdot \mathbb{1}(\hat{s}_{pre} \leq 1.5\max_{i\neq\hat{s}_{pre}} \delta_i)$.

This approach is applied in the real data analysis in Section 6, by which we obtain the estimation $\hat{s} = 1$, as seen in Figure 7.

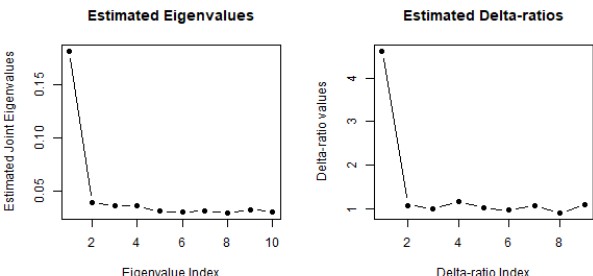

*Figure 7.* Estimation of the joint minimax eigenspace dimension for the real data application found in Section 6. Since $\delta_1 \geq 3\max_{i\neq 1} \delta_i$, we set $\hat{s} = 1$.

# D. Additional Simulation Results

In the present section, we provide additional numerical simulation comparisons between our proposed JIC-HD method and the HWGL, GLASSO and IPC-HD methods. We compare these methods in terms of average computational time, F-score, precision, recall, true positive rate (TPR), false positive rate (FPR).

## D.1. Computational Time

In this section, we provide comparisons for the computational time performance of our proposed JIC-HD method with methods found in the literature for hub detection in graphical models. The results in log-seconds for $p = 400$ are visualized in Figure 8

As we can observe from Figure 8, the method with the best computational time is the IPC-HD method, which has an average computational time below 1 second for all scenarios. The second best performing method is our JIC-HD, which remains between 10 seconds and 1 minute for all scenarios. Finally, the HWGL and GLASSO methods have the worst

performance in terms of computational time, remaining above 1 minutes for all scenarios with $p = 400$, and even having an average computational time above 1 hour for some cases.

Given the advantages in terms of common hub recovery illustrated in Figure 4, our JIC-HD method remains the best balance between computational time and common-hub recovery performance, even in high-dimensional scenarios.

## D.2. Measures of Hub-Recovery Performance

To deliver further insight into the performance of our model, we provide comparison of the empirical performance of our proposed JIC-HD and methods in the literature terms of F-score (Figure 9), precision (Figure 10), recall (Figure 11), true positive rate of common hub detection (Figure 12) and false positive rate (Figure 13).

As we can observe, across F-score, precision, recall, true positive rate, we observe that our method consistently obtains the best performance. This improved performance is especially notable when $p_C < p_I$, where the degrees of the common hubs are less than the degrees of the individual hubs. While other methods struggle to accurately recover this weaker signal, our method based on minimax optimization can filter out the individual signal and only recover the hubs that are common. Thus, our method is a reliable approach for the recovery of common hubs across multiple GGMs. In addition, observe that all methods obtain an FPR rate bounded by 0.05 in all scenarios. This was the result of the hub selection approach established in the simulation, which defines hubs as variables with connectivity above average connectivity plus two standard deviations.

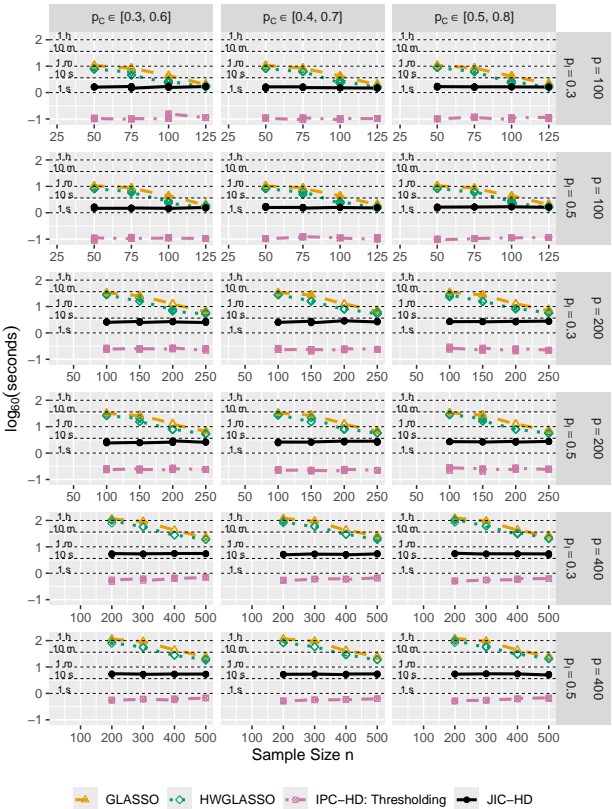

*Figure 8.* Comparison of the average computational time varying over $p$, $p_C$ and $p_I$. The averages are taken over 100 different replicates for each scenario. The JIC-HD methods provide the second best performance in terms of computational time.

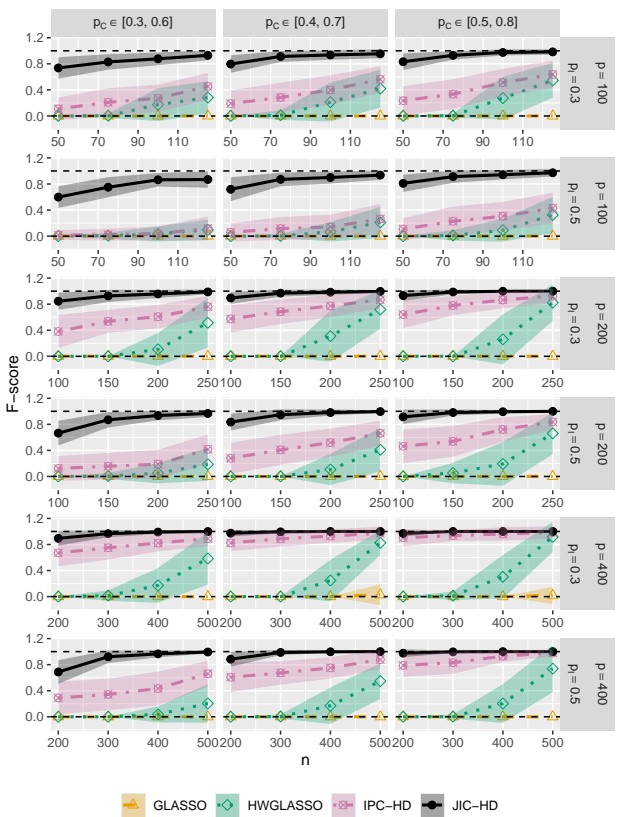

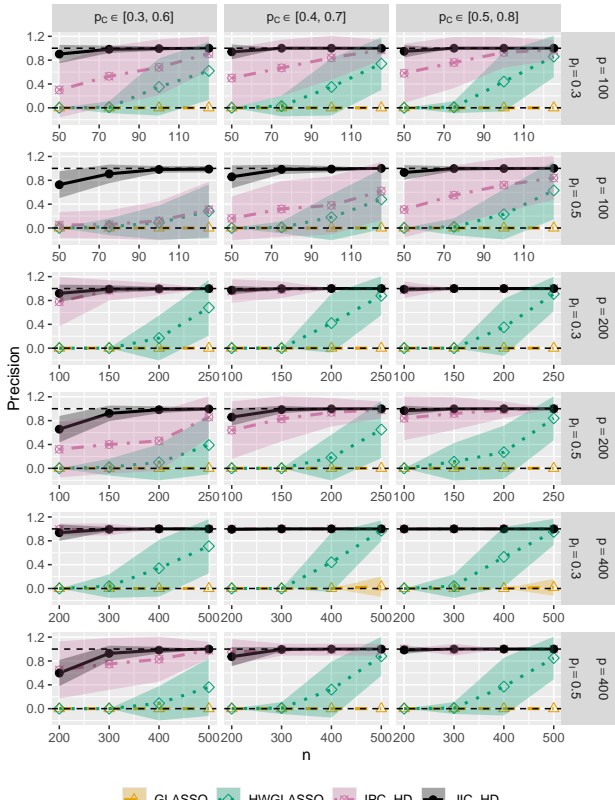

*Figure 9.* Visualization of the average F-score for our JIC-HD method compared to the GLASSO, HWGLASSO and IPC-HD method, for a variety of choices of $p$, $p_C$, $p_I$ and $n$. Our proposed JIC-HD method obtains the highest average F-score for all simulation scenarios. The largest gap is observed for $p_C < p_I$, *i.e.*, when the degree of the individual hubs is greater than the degrees of common hubs. Overall, we observe an advantage of our JIC-HD method for hub detection in terms of F-score.

*Figure 10.* Visualization of the average precision for our JIC-HD method compared to the GLASSO, HWGLASSO and IPC-HD method, for a variety of choices of $p$, $p_C$, $p_I$ and $n$. Our proposed JIC-HD method obtains the highest average precision for most simulation scenarios. The largest gap is observed for $p_C = 0.3 < p_I = 0.5$, *i.e.*, when the degree of the individual hubs is greater than the degrees of common hubs. Overall, we observe an advantage of our JIC-HD method for hub detection in terms of precision.

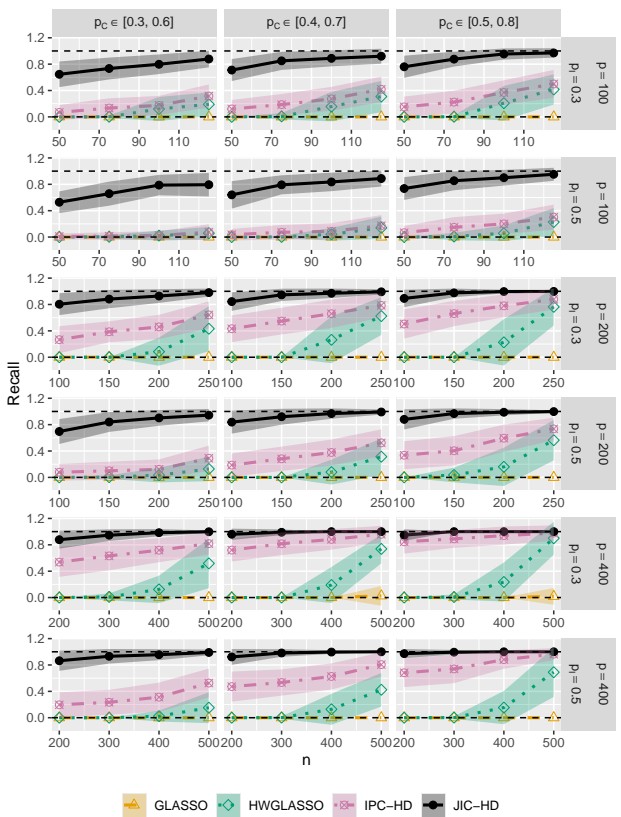

*Figure 11.* Visualization of the average recall for our JIC-HD method compared to the GLASSO, HWGLASSO and IPC-HD method, for a variety of choices of $p$, $p_C$, $p_I$ and $n$. Our proposed JIC-HD method obtains the highest average recall for all simulation scenarios. The largest gap is observed for $p_C < p_I$, *i.e.*, when the degree of the individual hubs is greater than the degrees of common hubs. Overall, we observe an advantage of our JIC-HD method for hub detection in terms of recall.

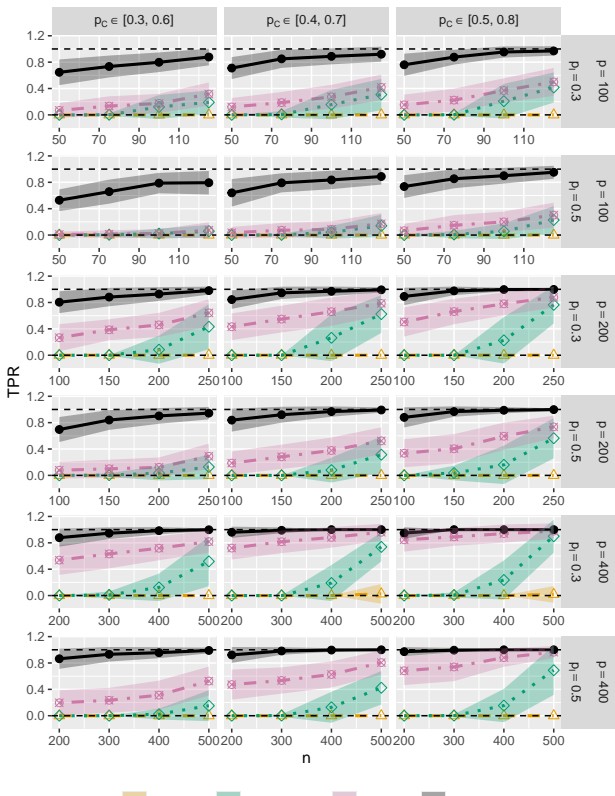

*Figure 12.* Visualization of the average true positive rate (TPR) for our JIC-HD method compared to the GLASSO, HWGLASSO and IPC-HD method, for a variety of choices of $p$, $p_C$, $p_I$ and $n$. Our proposed JIC-HD method obtains the highest average TPR for all simulation scenarios. The largest gap is observed for $p_C < p_I$, *i.e.*, when the degree of the individual hubs is greater than the degrees of common hubs. Overall, we observe an advantage of our JIC-HD method for hub detection in terms of TPR.

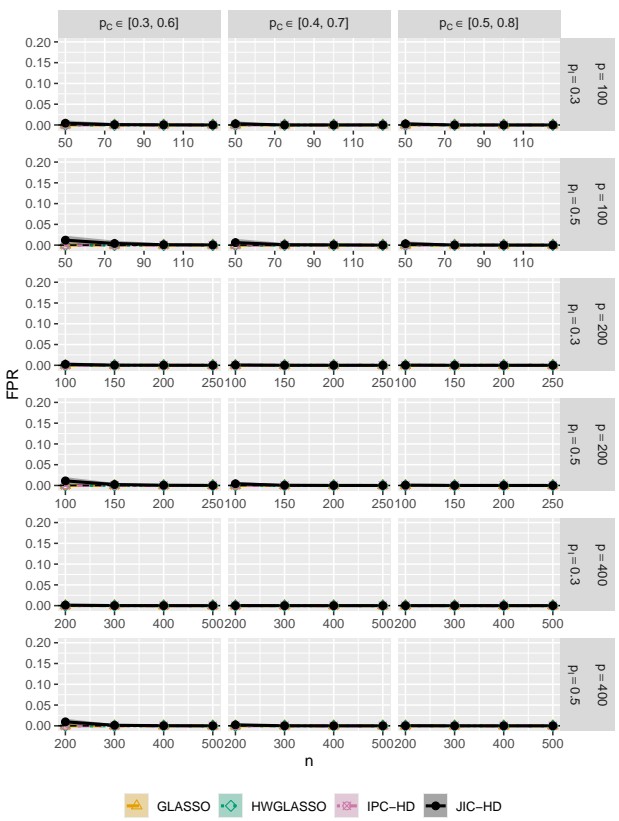

*Figure 13.* Visualization of the average false positive rate (FPR) for our JIC-HD method compared to the GLASSO, HWGLASSO and IPC-HD method, for a variety of choices of $p$, $p_C$, $p_I$ and $n$. We observe that all methods have an average FPR bounded by 0.05. Thus, FPR is not a major measurement of performance.

## D.3. Sensitivity Analysis of JIC-HD

Our proposed JIC-HD relies on the selection of the number of joint minimax eigenvectors $\widehat{s}$. In practice, we recommend $\widehat{s} = \sqrt{p}$, which we consider to be a reliable overestimation of the number of common hubs present in multiple populations. To explore the sensitivity of our method to the choice of $\widehat{s}$, we perform the following numerical experiments.

We follow the data generating process described in Section 5 of our main manuscript, with $K = 3$, $p \in \{200, 400\}$, $r \in \{5, 10, 15\}$, $p_C \in \{0.3, 0.4, 0.5\}$ and $p_I = 0.5$. We apply our proposed JIC-HD method, now varying the number of joint minimax eigenvectors $\widehat{s}$, along the following set of values: $\widehat{s} \in \{r, \sqrt{p}/2, \sqrt{p}, 3\sqrt{p}/2\}$. Here $\widehat{s} = r$ corresponds to an oracle estimator which uses the correct number of joint eigenvectors. The choices of $\widehat{s} = \sqrt{p}/2, \sqrt{p}, 3\sqrt{p}/2$ represent selections of $\widehat{s}$ that are agnostic to the underlying number of common hubs. We display the hub-recovery performance in terms of F-score in Figure 14.

As we observe from Figure 14, the JIC-HD obtains a high F-score for all selections of $\widehat{s}$. In particular, notice that the F-score becomes more favorable for higher values of $\widehat{s}$. For example, for $r = 5$, the lowest F-score is consistently attained by the choice of $\widehat{s} = s = 5$, while the most favorable performance is achieved by our JIC-HD with $\widehat{s} = 3\sqrt{p}/2 > 20$. On the other hand, for numerical simulations with number of common hubs $r = 15$, we have that the worse F-score is attained by the choice of $\widehat{s} = \sqrt{p}/2 < 10$, while the selection of JIC-HD with $\widehat{s} = s = 15$ and $\widehat{s} = 3\sqrt{p}/2 > 20$ attain a more desirable performance. Overall our JIC-HD attains desirable performance for all choices of $\widehat{s}$, and shows reliable common hub recovery for our suggested choice of $\widehat{s} = \sqrt{p}$.

## D.4. Comparison for Varying Number of Common Hubs

In Section 5 of our main manuscript, as well as Appendix D, we have analyzed the performance of our JIC-HD when the number of common hubs $r = 5$ is fixed. In the current section, we provide numerical simulation results for the detection of common hubs for $r \in \{5, 10, 15\}$. In Figure 15, visualize the F-score when the number of common hubs is $r = 10$, varying on $p$, $p_C$, $p_I$ and $n$. Additionally, in Figure 16, we visualize the F-score for simulations where $r = 15$. As observed, in all simulation scenarios explored, our proposed JIC-HD attains the best F-score, which confirms the superiority of our method for the task of common hub detection across multiple GGMs.

## E. Analysis of Lung Cancer Gene Expression Data

As discussed in Section 6 in the main paper, we estimate the presence of common hubs across two sub-types of lung cancer: lung adenocarcinoma (LUAD) and lung squamous cell carcinoma (LUSC). In the present appendix, we provide further visualization of the degrees of connectivity for the HWGLASSO and GLASSO. The results are visualized in Figure 17. As we observe, the GLASSO method does not detect any meaningful separation between hubs and non-hubs. On the other hand, the HWGLASSO detects the genes SFTPA1 and SFTPA2 as hub genes for both LUAD and LUSC cancer sub-types. This result is consistent with the common-hub genes detected with the JIC-HD method.

*Figure 14.* Visualization of the average F-score for our JIC-HD method for a variety of choices of $\widehat{s} = r, \sqrt{p}/2, \sqrt{p}, 3\sqrt{p}/2$. Notice that the highest F-score is consistently attained by the method with the highest choice of $\widehat{s}$. Overall, all choices of $\widehat{s}$ attain a desirable rate of common hub detection.

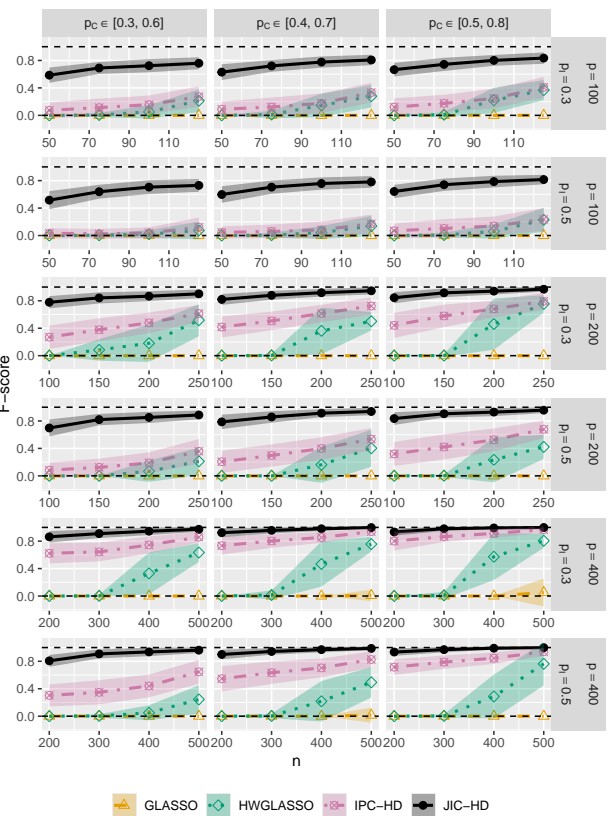

*Figure 15.* Visualization of the average F-score for our JIC-HD method compared to the GLASSO, HWGLASSO and IPC-HD method, for number of common hubs $r = 10$.

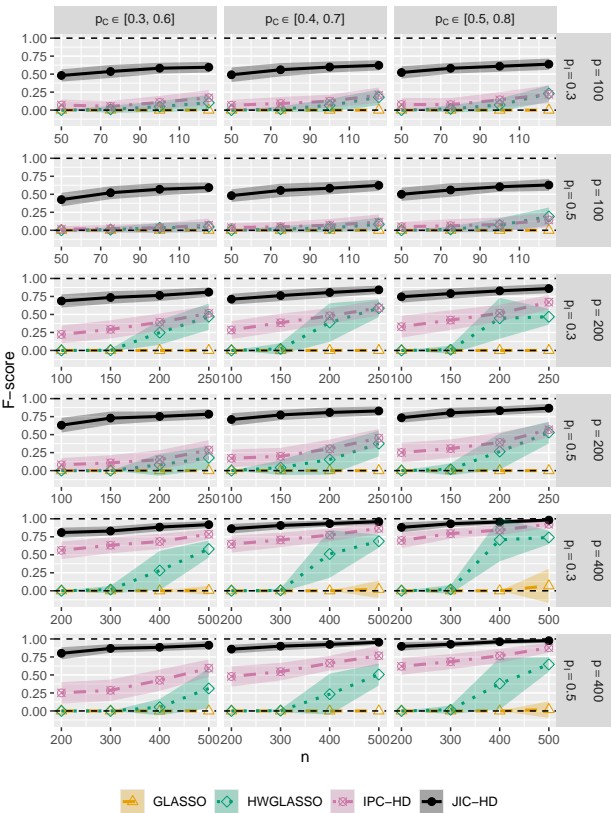

*Figure 16.* Visualization of the average F-score for our JIC-HD method compared to the GLASSO, HWGLASSO and IPC-HD method, for number of common hubs $r = 15$.

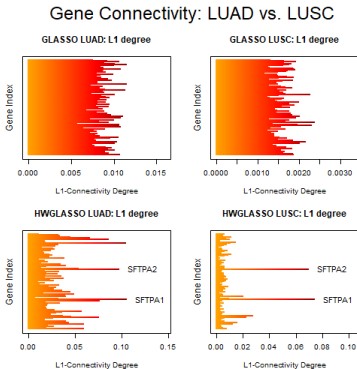

*Figure 17.* Comparison of $\ell_1$-connectivity degrees across the LUAD and LUSC cancer subtypes. We provide visualizations for both the GLASSO and HWGLASSO methods. Top panels: $\ell_1$ degrees of connectivity for the GLASSO method. No clear hub separation is detected, due to the uniform penalization. Center panels: $\ell_1$ degrees of connectivity for the HWGLASSO method. Only the genes SFTPA1 and SFTPA2 are detected as hubs common for both LUAD and LUSC sub-populations.

