# OpenReview forum: "Identifying Common Hubs in Multiple Gaussian Graphical Models"
_ICML.cc/2026/Conference — ICML 2026 regular_

### Official Review · Reviewer_BMHm · 2026-02-17

**Soundness:** 4
**Presentation:** 4
**Significance:** 3
**Originality:** 3
**Overall Recommendation:** 4
**Confidence:** 4

**Summary:**

This paper studies the problem of identifying common hub variables across multiple Gaussian graphical models corresponding to heterogeneous sub-populations. It proposes a new method, JIC-HD, which detects common hubs by estimating a minimax joint eigenspace of the covariance matrices, avoiding full graph estimation and remaining robust to large, non-sparse differences across networks. The authors establish theoretical guarantees for consistent common hub recovery under eigenspace decomposability assumptions. Simulation studies and a real gene-expression application demonstrate improved performance over existing hub-detection and joint graphical modeling methods, especially when individual hubs are stronger than common ones.

**Compliance With Llm Reviewing Policy:**

Affirmed.

**Final Justification:**

The author address my concerns, and I believe current score well represents the quality of the paper.

**Key Questions For Authors:**

1. Sensitivity to tuning parameter s: How sensitive is the method to the choice of the eigenspace dimension s? A robustness study could clarify whether performance degrades gracefully under misspecification.

2. Choice of threshold parameter κ: How should κ be selected in applications? Is there theoretical justification, prior literature support, or a data-driven selection strategy?

3. Realism of Condition 4: Under what practical scenarios is Condition 4 expected to hold? Can the authors provide intuition or examples where it may hold?

4. Baselines in simulations: Have the authors considered comparison methods that jointly estimate common and individual structures across graphs (rather than intersecting separately estimated hubs)?

**Limitations:**

yes

**Strengths And Weaknesses:**

Strengths
1. The paper studies a clearly defined and meaningful problem: identifying common hubs across multiple GGMs. The methodological motivation is well articulated and the logical development of the paper is coherent.
2. The proposed method is supported by theoretical guarantees, and the analysis appears rigorous. The work also shows a reasonable level of originality by connecting hub detection to a minimax joint eigenspace framework.

Weaknesses
1. The simulations do not examine sensitivity to the tuning parameter s. A robustness analysis would improve confidence in the stability of the proposed method.
2. The choice of the threshold parameter κ is not clearly justified. It would be helpful to provide supporting references or practical guidance for selecting this parameter.
3. Condition 4 appears strong, and it is unclear how realistic or verifiable it is in practice. A discussion of when this condition is likely to hold, and the consequences of its violation, would strengthen the paper.
4. Theorem 2 assumes that the minimax eigenspace dimension s is correctly specified. This requirement seems restrictive; some analysis of misspecification would improve the practical relevance of the theoretical results.
5. In the simulations, competing methods estimate hubs separately for each GGM and then intersect them. This comparison may favor the proposed approach. Including baselines that jointly recover common and individual structures across graphs would provide a fairer evaluation.

---

> ### Author Rebuttal · Authors · 2026-03-27
>
> We appreciate your thoughtful reviews on our manuscript. Please find our responses to your comments on the Weaknesses (W) and Questions (Q) below.
>
> 1. The choice of $s$ (W1,W4,Q1). Our theoretical guarantees are based on the true common eigenspace dimension $s$ to capture the essential ideas. The estimation of $s$ is further discussed in Supplementary Materials Section S3. For more general applications, we propose $\hat{s}=\sqrt{p}$ as an overestimation of $s$. It is motivated from the hub detection for a single population in [SMZ25], where such an overestimation remains theoretically valid. Intuitively, it suffices to ensure that the overestimated eigenspace can contain the hub-related signals, with negligible impacts from the non-hub noises. During the revision, we consider an extensive sensitivity analysis for $s=\sqrt{p}/2,\sqrt{p}, 3\sqrt{p}/2$. To provide further insights, we also explored the performance of our JIC-HD varying the number of common hubs $H_c =  5,10,15 $. Finally, for each hub variable, we allow the connection probability to vary as $p_{h} \sim Unif([0.3,0.6])$, introducing further heterogeneity to our hub signal.
>
>      Preliminary results on the effect of varying $\hat{s}$ on the true positive rate (TPR) and false positive rate (FPR) for $p = 400$ can be found in the file "525_Sens.pdf" from the anonymous github repository link:
>      https://anonymous.4open.science/r/ICML-2026-Reviewer-Response-Figures-6832/525_Sens.pdf
>      As we observe from the figure, in all our preliminary simulation scenarios, and for all choices $s=\{\sqrt{p}/2,\sqrt{p}, 3\sqrt{p}/2\}$, the TPR remains high, which signals the robustness of our proposed JIC-HD method to variable choices of $\hat{s}$. Additionally, we observe that the FPR remains near zero for all choices of $\hat{s}$ and $H_c$. This further confirms the robustness of our proposed JIC-HD method for the recovery of hubs.
>
> 2. The choice of $\kappa$ (W2,Q2). We recommend $\kappa=\mu+2\sigma$, where $\mu$ and $\sigma$ are the mean and standard deviation (SD) of the minimax influence measures $\omega(i)$'s (Line 253, right column). Our Theorem 2 suggests that $\kappa$ should be asymptotically no less than the estimation error $\max(p^{-\beta/4},\sqrt{\mathcal{E}(n,p)})$ of the minimax eigenspace in Theorem 1, and is smaller the better. The proof of Theorem 2 can also imply that our mean-plus-2SD approach for $\kappa$ satisfies $\kappa = O_{P}(\max(p^{-\beta/4},\sqrt{\mathcal{E}(n,p)}))$, which is asymptotically eligible and minimal. In addition, our numerical simulations confirm the reliability of this choice for the common hub detection.
>
> 3. Assumption 4 for covariance estimation (W3,Q3). Assumption 4 requires a standard estimation guarantee for the covariances $\Sigma^{(k)}$'s, which can be achieved by a wide variety of off-the-shelf covariance estimators. For example, the sample covariance estimator achieves $\mathcal{E}\propto\sqrt{p/n}$ [W19], while in high dimension, more structural covariance estimators typically achieve $\mathcal{E}\propto\sqrt{\log(p)/n}$ [BL08]. More discussions on covariance estimation will be included during revision.
>
> 4. Simulation baseline (W5,Q4). In our current simulations, we compare our proposed JIC-HD with the baselines: the GLASSO, hub-weighted GLASSO, and IPC-HD based on sub-population hub detections and intersection. We have not considered other methods for joint GGM estimation based on the following reasons: (1) Existing joint estimation methods are for the estimation of the precision matrices $\Theta^{(k)}$'s rather than the hub detection. (2) Existing joint estimation methods are based on the structural assumptions that may not be comparable to ours. For example, joint GGM estimation often requires the precision matrix differences $\Theta^{(k)} - \Theta^{(\ell)}$ to be sparse [ZMK25], but our settings and simulations break this assumption.
>
> During the revision of our manuscript, we will carefully revise according to your feedbacks. Thank you again for your valuable comments.
>
> References:
>
> [BL08] Bickel, P. J., and Levina, E. (2008). Covariance regularization by thresholding. The Annals of Statistics, 36, 2577–2604.
>
> [ZMK25] Zhao, B., Ma, C., & Kolar, M. (2025). Trans-glasso: A transfer learning approach to precision matrix estimation. Journal of the American Statistical Association, (just-accepted), 1-21.
>
> [SMZ25] Sánchez Gómez, J. Á., Mo, W., Zhao, J., & Liu, Y. (2025). Hub Detection in Gaussian Graphical Models. Journal of the American Statistical Association, 120(552), 2397-2409.
>
> [W19] Wainwright, M. J. (2019). High-dimensional statistics: A non-asymptotic viewpoint (Vol. 48). Cambridge University Press.

---

> > ### Author Rebuttal · Reviewer_BMHm · 2026-04-02
> >
> > After reading the rebuttal, I have no further questions. Thank you.

---

> > > ### Author Response · Authors · 2026-04-08
> > >
> > > We appreciate your thoughtful review on our manuscript, and we believe that your concerns have been adequately addressed. We would also like to highlight that we consider our novel minimax eigenspace problem and its success in common hub detection as a significant contribution to the state-of-the-art multiple network research. Our theoretical justifications are based on weaker structural model assumptions compared to the state-of-the-art's, while our common hub detection guarantees are also new. The extensive simulations and real-data application also justify the empirical success of our method in fairly general data generating processes and various settings.
> > >
> > > In light of your acknowledgement, we hope that you could consider to recommend a higher score to our submission. We deeply appreciate your feedbacks, and will ensure that your observations and recommendations are reflected in the final version of the manuscript.

---

### Official Review · Reviewer_Z26s · 2026-03-07

**Soundness:** 1
**Presentation:** 2
**Significance:** 2
**Originality:** 2
**Overall Recommendation:** 2
**Confidence:** 3

**Summary:**

This paper proposes JIC-HD, a spectral method to detect common hubs across K Gaussian graphical models without explicitly estimating each subpopulation precision matrix. The paper defines the common hub set as the intersection $J := \cap_{k=1}^K H_k$ and introduces a minimax joint eigenspace estimator. The authors provide theory controlling the estimation error of the minimax eigenspace (Theorem 1) and a hub recovery guarantee that bounds false inclusions (Theorem 2). Empirically, the method is compared to GLASSO/HWGLASSO/IPC-HD in simulations and applied to lung gene expression data.

**Compliance With Llm Reviewing Policy:**

Affirmed.

**Key Questions For Authors:**

1) The algorithm requires setting $\hat{s}$ and recommends $\hat{s}=\sqrt{p}$, but Theorems 1–2 are stated for $s$ and even assume correct specification. Can the theory be extended to the actual procedure with $\hat{s}$?
2) Theorem 2 only shows $\hat{J}(\kappa)\subseteq J$. Is it possible to provide conditions guaranteeing recovery of all common hubs, or at least a lower bound on true positive rate?
3) In simulations, the number of common hubs is fixed to 5, and $\hat{s}$ is fixed as $[\sqrt{p}] $. Please add experiments varying (i) number of common hubs (e.g., 10/20/40), and (ii) $\hat{s}$, to demonstrate robustness.
4) In the real data application, what exact $\hat{s}$ and $\kappa$ were used, and how sensitive are the detected hubs to these choices? Showing sensitivity curves would strengthen credibility.

**Limitations:**

yes

**Strengths And Weaknesses:**

Soundness
1) Hub definition is nonstandard and insufficiently justified.
Hubs are typically defined via degree or support-level connectivity in the graph, but the paper defines hubs using a column norm criterion:
$\min_{h\in H}\|\Theta_{\cdot h}\|2^2 \gg \max_{i\notin H} \|Theta_{\cdot i}\|_2^2$.
This is a major modeling choice, which is not standard. Authors should explain more why this notion is justified.
2) A3-1 seems too strong.
A3-1 states that $\mathrm{span}(V_{J,k})$ is close across $k$ in chordal distance. Since the algorithm’s success relies on recovering the common component via a joint eigenspace, this assumption reads as “the common eigenspace is already well aligned,” which is close to the desired conclusion. It would be helpful if authors provide sufficient conditions under which A3-1 hold.
3) Hub and spiked eigenvalue connection is not clear.
The paper motivates its spectral approach by suggesting that the presence of hub nodes leads to spiked eigenvalues and a dominant leading eigenspace. However, the manuscript does not show why the proposed hub definition actually implies such a spiked spectral structure.
4) Connection between common hubs and a shared low-rank structure needs clarification.
The paper states that the presence of common hubs induces a shared low-rank structure.
But conceptually, low rank seems to be driven by few hub nodes rather than “commonness”. The manuscript should clearly separate: Is low-rank induced because hubs are common across graphs, or because the number of common hubs is sparse?
5) $\hat{s}$ v.s. $s$.
Algorithmically, the method requires choosing $\hat{s} \geq s$, and recommends $\hat{s}=\sqrt{p}$ based on expecting $r=O(\sqrt{p})$.
But the theoretical results are stated only for $s$, not $\hat{s}$. This makes it hard to interpret what the theorems imply about the actual algorithm as implemented with $\hat{s}$.
6) Theorem 2 is not selection consistency.
Theorem 2 establishes $\hat{J}(\kappa) \subseteq J$ with high probability under conditions and thresholding.
This controls false positives but gives no guarantee on recovering all true common hubs (true positive rates), which is a substantial gap given the paper’s goal of “recovering common hubs.”

Presentation
The paper is readable at a high level, but key modeling and logical transitions need clearer exposition
1) The jump from hubness defined via $\ell_2$ norms to spiked eigenstructure and then to common eigenspace alignment needs more motivation and explanation.
2) The role of $\hat{s}$ in the algorithm and how it is handled in theory should be more explicit.

Significance
Detecting common hubs across heterogeneous GGMs is an important problem, and the idea of using a minimax aggregation to reduce the influence of individual hubs is conceptually interesting. However, significance is limited by (i) too strong assumption A3-1 and (ii) little information on when the algorithm works well and not.

Originality
The joint minimax eigenspace formulation and the accompanying joint influence measure are the most novel aspects of the paper. Hence, the theoretical justification appears to hinge critically on a strong common-eigenspace condition (A3-1), which risks narrowing the regime where the guarantees apply. Beyond providing simulations, it would strengthen the contribution to more explicitly explain and characterize the settings in which the existing strategies fail while the proposed minimax-based approach still succeeds, and to discuss robustness when the eigenspace alignment assumption is violated.

---

> ### Author Rebuttal · Authors · 2026-03-30
>
> We appreciate your thoughtful reviews on our manuscript. Please find our responses to your comments on the Soundedness (Sn), Presentation (P), Significance (Sg), Originality (O) and Questions (Q) below.
>
> 1. Hub and spiked eigenspace (Sn1,Sn3,P1). We define hubs from non-hubs based on their connectivity in asymptotically different orders. As long as the non-zero entries of $\Theta$ are bounded, the hubs defined from the column-wise $\ell_{2}$-norm and the column-wise $\ell_{0}$-norm are equivalent. Furthermore, under regularity conditions, it has been proven that the presence of $\ell_2$-hubs in a precision matrix $\Theta$ implies a spiked eigenstructure of $\Theta$ [SMZ25].  Therefore, low-rank structures can be exploited for hub detection. Figure 1 in our manuscript illustrates this, where $\Theta_{J},\Theta_{I1}$ are the spiked components of $\Theta_{1}$. Intuitively, the dense connections in two rows and columns of $\Theta_{1}$ are contributing the rank-2 components with the eigenvalues in $O(p)$, while the eigenvalues of the remainder $\tilde{\Theta}_{R1}$ are in $O(1)$.
>
> 2. Common hub and common eigenspace (Sn2,Sn4,Sg1,O1). A3-1 is NOT a restrictive model assumption, but a consequence of the presence of common hubs under fairly general regularity conditions. As you have pointed out in Sn4, the spiked eigenstructure of each $\Theta^{(k)}$ is driven by the presence of hubs. We can further establish that under certain regularity conditions, the presence of common hubs implies that the corresponding spiked eigenspaces are well aligned as in A3-1. Figure 1 illustrates such a case, since the common component $\Theta_{J}$ induces a shared low-rank structure on $\Theta_1,\Theta_2$. Assumption 3 is a concise set of general regularity conditions, based on which our theoretical guarantees are established. These can be verified under fairly general sufficient conditions, such as the presence of common hubs but absence of individual hubs, or more generally, that the common and individual hubs appear in separable communities within the networks. More general sufficient conditions will be included in our revised manuscript.
>
> 3. Choice of $s$ (Sn5,P2,Q1,Q4). Our theoretical guarantees are based on the true common eigenspace dimension $s$ to capture the essential ideas. The estimation of $s$ for our real data application is provided in Supplementary Materials Section S3. For more general applications, we propose $\hat{s}=\sqrt{p}$ as an overestimation of $s$. It is motivated from the hub detection for a single population in [SMZ25], where such an overestimation remains theoretically valid. Intuitively, it suffices to ensure that the overestimated eigenspace can contain the hub-related signals, with negligible impacts from the non-hub noises. Our simulations further confirm the reliability of $\hat{s}=\sqrt{p}$ for the common hub detection.
>
> 4. Choice of $\kappa$ (Q4). We recommend $\kappa=\mu+2\sigma$, where $\mu$ and $\sigma$ are the mean and standard deviation (SD) of the minimax influence measures $\omega(i)$'s (Line 253, right column). Our Theorem 2 suggests that $\kappa$ should be asymptotically no less than the estimation error $\max(p^{-\beta/4},\sqrt{\mathcal{E}(n,p)})$ of the minimax eigenspace in Theorem 1, and is smaller the better. The proof of Theorem 2 can also imply that our mean-plus-2SD approach for $\kappa$ satisfies $\kappa = O_{P}(\max(p^{-\beta/4},\sqrt{\mathcal{E}(n,p)}))$, which is asymptotically eligible and minimal. In addition, our numerical simulations confirm the reliability of this choice for the common hub detection.
>
> 5. Common hub detection guarantee (Sn6,Q2). While our Theorem 2 establishes $\hat J \subset J$ for generality, there are reasonable sufficient conditions where we could further establish $\hat J = J$, such as the presence of common hubs but absence of individual hubs, or more generally, that the common and individual hubs appear in separable communities within the networks. More discussions will be included in our revised manuscript.
>
> 6. More experiments for robustness (Sg2,Q3,O2). We have run preliminary simulations that confirm that our proposed JIC-HD outperforms the baselines under various number of common hubs $r=5,10,15$ and varying magnitudes in the common hubs ("525_Comp.pdf"). Additional preliminary simulations confirm our JIC-HD has high common-hub recovery, robust to variation of $\hat{s}$ ("526_Sens.pdf"). These preliminary results can be found in the anonymous github link: https://anonymous.4open.science/r/ICML-2026-Reviewer-Response-Figures-6832/. Our revised manuscript will include full-scale simulations and further discussions.
>
> During the revision of our manuscript, we will carefully revise according to your feedbacks. Thank you again for your valuable comments.
>
> References:
> [SMZ25] Sánchez Gómez, J. Á., Mo, W., Zhao, J., & Liu, Y. (2025). Hub Detection in Gaussian Graphical Models. Journal of the American Statistical Association, 120(552), 2397-2409.

---

> > ### Author Rebuttal · Reviewer_Z26s · 2026-04-01
> >
> > Thank for the response. However, I still have some concerns.
> > (1) On the hub definition. While we appreciate the reference, the equivalence between ℓ2-based and degree-based hubs appears to rely on additional assumptions on the magnitudes of nonzero entries (e.g., not vanishing and relatively homogeneous). Such conditions seem restrictive and are not explicitly stated or justified in the manuscript. Therefore, the concern that the proposed hub definition deviates from the standard notion is not fully resolved.
> > (2) On Assumption A3-1. We acknowledge that in certain structured settings, common hubs may induce aligned eigenspaces. However, this does not hold in general. For instance, even if two graphs share the same common hub set, differences in the strength or distribution of connections can lead to substantially different leading eigenspaces. Hence, A3-1 does not follow as a consequence of common hubs alone, but instead imposes a strong structural requirement (essentially requiring a shared dominant eigenspace component). This reinforces the concern that the assumption encodes a key part of the desired conclusion.
> > (3) On the role of (\hat{s}). Our question was not about the practical choice of (\hat{s}), but whether the theoretical guarantees extend to the case (\hat{s} \ge s). In particular, it is unclear whether the results established for the correctly specified dimension (s) continue to hold under overestimation. The reference [SMZ25] considers a single GGM, whereas the present setting involves multiple graphs and a minimax eigenspace, so it is not evident that the same argument directly applies. This theoretical gap remains unaddressed.
> > (5) On Theorem 2 and sufficient conditions. Unfortunately, I am not convinced that Theorem 2 is fully justified in the current version. In particular, without explicit sufficient conditions under which the assumptions hold, it is unclear whether the theorem is applicable in general settings. This concern affects my overall assessment.

---

> > > ### Author Response · Authors · 2026-04-08
> > >
> > > Thank you for your additional comments.
> > >
> > > 1. **Hub definition.** To clarify, we do not intend to argue the equivalence of the $\ell_{0}$-and $\ell_{2}$-based definitions. Our $\ell_{2}$-definition has its own advantages in accounting for the connection strengths. In a GGM, the entries in the precision matrix $\Theta$ represents the strength of partial correlations. The $\ell_{2}$-definition can take the strength of relationship into account while the $\ell_{0}$-definition may ignore such information. In particular, it has been shown that an $\ell_{2}$-based hub variable is strongly related to the remaining variables in terms of mutual information [SMZ25].
> > >
> > > 2. **Sufficient condition for A3-1.** We will make the structural assumptions clear in a sufficient condition A5 below. Write $\Theta^{(k)} = \Theta_{J}^{(k)} + \Theta_{I}^{(k)} + \Theta_{R}^{(k)}$, where the nonzero entries of $\Theta_{J}^{(k)}$ are those in $\Theta^{(k)}$ with rows or columns in $J$, and the nonzero entries of $\Theta_{I}^{(k)}$ are those in $\Theta^{(k)}-\Theta_{J}^{(k)}$ with rows or columns in $I^{k}\backslash J$. For every variable $i$, define its neighborhood $N_{i}^{(k)} = \\{j:\Theta_{ij}^{(k)}\neq 0\\}$. Define the common hub neighborhood $N_{J}^{(k)}=\bigcup_{j\in J}N_{j}^{(k)}$ and individual hub neighborhood $N_{I}^{(k)}=\bigcup_{i\in I^{k}}N_{i}^{(k)}$. **(A5) (i) $\Theta_{J}^{(k)}\equiv\Theta_{J}$,  (ii) $N_{J}^{(k)}\cap N_{I}^{(k)}=\emptyset$, (iii) $\| \Theta_{R}^{(k)} - \Theta_{R}^{(l)} \|_{2} = O(p^{-\beta/4})$ uniformly for $k,l$'s.** In particular, **A1, A2, A5 together imply A3-1.**
> > >
> > > A5(i) underlies the eigenspace alignment in A3-1. It is motivated from the structure sharing in multiple GGM estimation [ZMK25].  It can be relaxed such that $\Theta_{R}^{(k)}$ incorporates $\Theta_{J}^{(k)}-\Theta_{J}$ and (iii) remains to hold. For A5(ii), we consider $N_{J}^{(k)}$ and $N_{I}^{(k)}$ as disjoint communities within the network of $\Theta^{(k)}$. It holds trivially if $I^{k}=\emptyset$ across $k$'s. More generally, it can be relaxed such that $\Theta_{R}^{(k)}$ incorporates the connections in $N_{J}^{(k)}\cap N_{I}^{(k)}$ and (iii) remains to hold. A5(iii) requires that the cross-population differences are small in the spectral norm, which could be substantially weaker than those in joint GGM estimation [ZMK25] that require sparsity among the differences.
> > >
> > > While our theoretical justifications rely on A3-1, our method can be successful in more generic settings. Our simulations consider the generic form $\Theta^{(k)} = \Theta_{J}^{(k)} + \Theta_{I}^{(k)} + \Theta_{R}^{(k)}$, whose nonzero entries and magnitudes are generated independently for each component. While our DGP does not follow A5 above, and A3-1 may not hold, our numerical results confirm the outperformance of the proposed JIC-HD under various settings.
> > >
> > > 3. **The overestimation of $\hat{s}$.** Our current theoretical guarantees are not yet extended to $\hat{s}\ge s$ due to the technical challenge of characterizing the minimax eigenspace problem in such a scenario. Nevertheless, our theoretical insight is built upon the theory in a single population [SMZ25], where minimizing the explained variance with $\hat{s}\ge s$ should approximately result in $span(V_{\widehat{s}})\supseteq span(V_{s})$, which will suffice for the guarantee in Theorem 2. A similar strategy has also been proven successful for multi-source eigenspace analysis in [STG25]. Our simulation study confirms such a phenomenon, and we aim to fill the theoretical gap in our future work.
> > >
> > > 4. **On $\hat{J}=J$ in Theorem 2.** To be specific, **under the same set of conditions in Theorem 2, and in addition, the above A5, the probabilistic statement in Theorem 2 holds with $\hat{J}=J$**. For general applications, our guarantee $\hat{J}\subseteq J$ could be of useful scientific interest, especially when shortlisting common hub candidates is helpful. Moreover, the separate-detect-then-intersect strategy as our comparison benchmark mainly suffers from missing the true common hubs due to multiplying the detection rates, and our method is guaranteed with no missing by Theorem 2.
> > >
> > > Despite the aforementioned theoretical limitations, we consider our novel minimax eigenspace problem and its success in common hub detection as a significant contribution to the state-of-the-art multiple network research. Existing methods may rely on even stronger structural assumptions for joint network estimation [ZMK25], or will suffer from the discounted detection rates in intersecting set estimates. Our numerical results under fairly general DGP also confirm the efficiency and robustness of the proposed method. Therefore, we would hope your reconsideration of our research and the recommended scores. Thank you again for the thoughtful review and discussion.
> > >
> > > [STG25] Sergazinov, R., Taeb, A., and Gaynanova, I. (2025). A spectral method for multi-view subspace learning using the product of projections. Biometrika, to appear.

---

### Official Review · Reviewer_6DFh · 2026-03-11

**Soundness:** 3
**Presentation:** 3
**Significance:** 3
**Originality:** 3
**Overall Recommendation:** 4
**Confidence:** 3

**Summary:**

The paper proposes a new method for common hub detections in Gaussian Graphical Models by thresholding the Euclidean norm of top $s$ leading eigenvectors that minimize the maximal total explained variance in a set of networks. The intuition is based on the low-rank structure of aggregated eigenspace, and theoretical guarantees depend on model linearity and orthogonal decompositions.

**Compliance With Llm Reviewing Policy:**

Affirmed.

**Final Justification:**

My question is resolved.

**Key Questions For Authors:**

1. How robust is the proposed method in discovering common hubs under varying signals?
2. How dependent is current method on the linearity, especially the eigen-decomposition metric?
3. Manifold gradient descent method is also costly thoughthe current method avoids precision matrix estimations. I'm curious about whether there'd be obvious computational savings by bypassing the precision matrix estimation?

I'm willing to increase my score if authors can help address my questions and concerned limitations.

**Limitations:**

- High dependency on the linearity.
- Possible trivial detections when signal is moderate.

**Strengths And Weaknesses:**

Strengths:

- The paper is well-written and easy to follow. I like the paper's presentation where intuitions are demonstrated to help understand the method design.
- The proposed method avoid the computationally expensive precision matrix estimation in high dimension.
- Theoretical justification seems mostly correct.

 Weaknesses:
- Only true positive is concerned. The theoretical guarantee is placed to show that the detected common hubs are contained in real common hubs. However, it is concerning that zero detection is possible under the moderate signal. Real data experiment also supports a small number of detections.
- Missing experiment to support the exclusion of any false detections.
- The linearity of GGM and eigen-decomposition required for the proposed method seems to be highly correlated, but discussion on the linearity dependence is missing for general applications.
- Typo at sec. 3 (line 252) $h\in\mathcal{P}$ should be $i\in\mathcal{P}$.

---

> ### Author Rebuttal · Authors · 2026-03-30
>
> We appreciate your thoughtful reviews on our manuscript. Please find our responses to your comments on the Weaknesses (W), Questions (Q) and Limitations (L) below.
>
> 1. Linearity assumption (W3,Q2,L1): Our proposed procedure for common hub detection relies on two main assertions of linearity: (1) the presence of hubs in a precision matrix $\Theta$ induces a low-rank structure in $\Theta$; and (2) when multiple precision matrices contain common hubs, we expect a shared low-rank structure. As we discuss below, these two assertions are not restrictive assumptions, but instead general verifiable conditions. We will include these vital discussions in our revised mansucript.
>
>      (1) Under certain regularity conditions, it has been proven that the presence of hubs in a precision matrix $\Theta$ implies a spiked eigenstructure of $\Theta$ [SMZ25].  Therefore, low-rank structures can be exploited for hub detection. Figure 1 in our manuscript illustrates this, where $\Theta_{J},\Theta_{I1}$ are the spiked components of $\Theta_{1}$. Intuitively, the dense connections in two rows and columns of $\Theta_{1}$ are contributing the rank-2 components with the eigenvalues in $O(p)$, while the eigenvalues of the remainder $\tilde{\Theta}_{R1}$ are in $O(1)$.
>
>      (2) The shared spiked eigen-space across precision matrices described in Assumption 3 is NOT a restrictive model assumption, but a consequence of the presence of common hubs. As mentioned in (1), the presence of hubs in each sub-population implies a spiked eigenstructure [SMZ25]. The presence of common hubs among sub-populations further implies the common spiked eigenspace as in Assumption 3. This is illustrated in Figure 1. Intuitively, the components $\Theta_{J}+\Theta_{I1},\Theta_{J}+\Theta_{I2}$ have eigenvalues in $O(p)$, while the eigenvalues of the remainders $\Theta_{R1},\Theta_{R2}$ are in $O(1)$. The common component $\Theta_{J}$ across $\Theta_{1},\Theta_{2}$ contributes a shared low rank structure for both matrices. In our manuscript, we introduce Assumption 3 as a concise set of regularity conditions, based on which our theoretical guarantees are established. These can be verified under fairly general sufficient conditions, such as the presence of common hubs but absence of individual hubs, or more generally, that the common and individual hubs appear in separable communities within the networks.
>
> 3. Common hub detection guarantee (W1,L2). While our Theorem 2 establishes $\hat J \subset J$ for generality, there are reasonable sufficient conditions where we could further establish $\hat J = J$, such as the presence of common hubs but absence of individual hubs, or more generally, that the common and individual hubs appear in separable communities within the networks. Furthermore, our experiment results confirm that our method enjoys consistently low false positive rates in all scenarios (Point 4 below). For the real-data application, the detailed analysis in Supplementary Materials Section S5 suggests that, the LUSC cancer subtypes may only contain the discovered two hub genes, confirming the reliability of the discovery.
>
> 4. Experiment results for false detections and varying signals (W2,L2,Q1). We have provided extensive metrics on the F-score, precision, recall, and false positive rate (FPR) in our Supplementary Materials Section S4.2, which supports that our method can also exclude false/trivial detections. In particular, our proposed JIC-HD achieves the best true common hub detection performance, while the false common hub detection (FPR) is low (<= 0.01 in most scenarios). To provide a more extensive analysis, we have considered preliminary simulations with varying numbers of common hubs $r=5,10,15$ and varying magnitudes in the common hubs, where our JIC-HD method remains to be the best consistently. Full-scale simulation experiments will be performed in our revised manuscript, and more estimation scenarios will be considered.
>
> 5. Optimization performance (Q3). For our manifold gradient descent method, we compare its computational time with other baselines in Supplementary Materials Section S4.1. Our JIC-HD based on manifold gradient descent has substantial computational advantages (in <= 30 seconds) over the GLASSO and HWGLASSO (in minutes) that rely on precision matrix estimation. For example, the GLASSO and HWGLASSO consume >= 35 minutes for $(p,n)=(400,200)$. Another baseline IPC-HD can also avoid the precision matrix estimation and enjoy similar computational advantages, but its statistical performance is poor in other comparisons.
>
> During the revision of our manuscript, we will carefully revise according to your feedbacks, and make sure that the compiled manuscript will not contain typos (such as W4). Thank you again for your valuable comments.
>
> References:
>
>
> [SMZ25] Sánchez Gómez, J. Á., Mo, W., Zhao, J., & Liu, Y. (2025). Hub Detection in Gaussian Graphical Models. Journal of the American Statistical Association, 120(552), 2397-2409.

---

### Official Review · Reviewer_H72f · 2026-03-11

**Soundness:** 3
**Presentation:** 4
**Significance:** 3
**Originality:** 3
**Overall Recommendation:** 4
**Confidence:** 1

**Summary:**

This paper proposes JIC-HD, which introduces a joint minimax feature space to identify common hubs across multiple GGMs without explicitly estimating all subpopulation GGMs. Both theoretical analysis and empirical studies demonstrate the effectiveness of the proposed method.

**Compliance With Llm Reviewing Policy:**

Affirmed.

**Final Justification:**

The authors propose the joint inverse components for hub detection method. The effectiveness and advantages of the method have been validated by numerical simulations and real datasets. Overall, the paper is well written and somewhat innovative.

**Key Questions For Authors:**

see weaknesses

**Limitations:**

yes

**Strengths And Weaknesses:**

I apologize that I am not an expert in this field. Despite my best efforts to carefully read, the following comments reflect my limited understanding and are offered with caution.

Strengths:
1. The authors introduce a joint minimax feature space into the public GGM identification method, overcoming the limitation of requiring a subpopulation GGM to identify public hubs.
2. The paper does not remain purely theoretical. Instead, the authors make a commendable effort to evaluate the proposed method using real-world datasets.
3. The paper is well written. Even though I am not an expert in this field, the background and motivation are presented clearly enough to be relatively easy to follow.

Weaknesses:
I apologize that I was unable to fully understand these formulas, and therefore I may not be able to provide accurate comments. There is a problem with the font in the paper, perhaps due to the compiler.

---

> ### Author Rebuttal · Authors · 2026-03-27
>
> We appreciate your careful review on our manuscript.  We would like to summarize our key contributions below.
>
> 1. We propose the JIC-HD method for the detection of common hubs across multiple Gaussian graphical models (GGMs). Our method enjoys desirable theoretical guarantees of common hub recovery, and improvements in terms of hub recovery and computational complexity when compared to other methods. We confirm the usefulness of our proposed JIC-HD for recovering hubs on multi-population cancer gene-expression data.
>
> 2. To our best knowledge, we provide the first method for the recovery of common hubs across multiple GGMs that bypasses the full estimation of all GGMs, and exploits the common structure shared by the GGMs. Thus the JIC-HD enjoys state of the art hub recovery and computational performance.
>
> 3. Our method is based on a novel notion of joint minimax eigenspaces for multiple matrices, which requires advanced optimization techniques under a non-convex smooth manifold constraint. We provide extensive discussions on motivation, algorithm, assumptions, and best practices on the choices of tuning parameters.
>
> During the revision of our manuscript, we will carefully review and make sure that the final manuscript will not contain typos, or font problems. Thank you again for your comments.

---

> > ### Author Rebuttal · Reviewer_H72f · 2026-04-01
> >
> > I thank the authors for the detailed response. I have no further questions at this stage.

---

> > > ### Author Response · Authors · 2026-04-08
> > >
> > > We appreciate your careful review and discussion on the strengths of our research. In light of your acknowledgement, we hope that you could consider to recommend a higher score to our submission. We deeply appreciate your feedbacks, and will ensure that your observations and recommendations are reflected in the final version of the manuscript.

---

### Decision · Program_Chairs · 2026-04-30

**Decision:**

Accept (regular)

**Comment:**

Learning and identifying hub nodes in graphical models is an important problem with many applications, for example in identifying genetic regulators from expression data as considered in the present submission. The authors consider a setting with multiple environments, each with their own graph, and the goal is to identify the common hubs between these graphs. The authors translate this into a low-rank signal detection problem, and solve it from this perspective. The contributions are remarkably complete: A framework, methodology, theory, and both simulated and real data experiments are all covered.

Reviewers praised the completeness of the results, and in particular the real data application. The main criticism came from a single reviewer who raised concerns about the assumptions. These concerns would be more pressing if the submission lacked empirical validation. While I encourage the authors to provide more discussion on the limitations of the assumptions in the camera ready, this does not diminish the overall scope of the contributions.